# Stochastic Recursive Gradient Descent Ascent for Stochastic Nonconvex-Strongly-Concave Minimax Problems

**Luo Luo**[1]    **Haishan Ye**[2]    **Zhichao Huang**[1]    **Tong Zhang**[1]

[1]Department of Mathematics, The Hong Kong University of Science and Technology
[2]Shenzhen Research Institute of Big Data, The Chinese University of Hong Kong, Shenzhen

luoluo@ust.hk    hsye_cs@outlook.com    zhuangbx@connect.ust.hk    tongzhang@ust.hk

## Abstract

We consider nonconvex-concave minimax optimization problems of the form $\min_{\mathbf{x}} \max_{\mathbf{y} \in \mathcal{Y}} f(\mathbf{x}, \mathbf{y})$, where $f$ is strongly-concave in $\mathbf{y}$ but possibly nonconvex in $\mathbf{x}$ and $\mathcal{Y}$ is a convex and compact set. We focus on the stochastic setting, where we can only access an unbiased stochastic gradient estimate of $f$ at each iteration. This formulation includes many machine learning applications as special cases such as robust optimization and adversary training. We are interested in finding an $\mathcal{O}(\varepsilon)$-stationary point of the function $\Phi(\cdot) = \max_{\mathbf{y} \in \mathcal{Y}} f(\cdot, \mathbf{y})$. The most popular algorithm to solve this problem is stochastic gradient decent ascent, which requires $\mathcal{O}(\kappa^3 \varepsilon^{-4})$ stochastic gradient evaluations, where $\kappa$ is the condition number. In this paper, we propose a novel method called Stochastic Recursive gradiEnt Descent Ascent (SREDA), which estimates gradients more efficiently using variance reduction. This method achieves the best known stochastic gradient complexity of $\mathcal{O}(\kappa^3 \varepsilon^{-3})$, and its dependency on $\varepsilon$ is optimal for this problem.

## 1  Introduction

This paper considers the following minimax optimization problem

$$\min_{\mathbf{x} \in \mathbb{R}^d} \max_{\mathbf{y} \in \mathcal{Y}} f(\mathbf{x}, \mathbf{y}) \triangleq \mathbb{E}\left[F(\mathbf{x}, \mathbf{y}; \boldsymbol{\xi})\right], \tag{1}$$

where the stochastic component $F(\mathbf{x}, \mathbf{y}; \boldsymbol{\xi})$, indexed by some random vector $\boldsymbol{\xi}$, is $\ell$-gradient Lipschitz on average. This minimax optimization formulation includes many machine learning applications such as regularized empirical risk minimization [41, 52], AUC maximization [39, 48], robust optimization [14, 46], adversarial training [16, 17, 40] and reinforcement learning [13, 43]. Many existing work [8, 9, 12, 13, 18, 28, 33, 34, 41, 45, 48, 50, 52] focused on the convex-concave case of problem (1), where $f$ is convex in $\mathbf{x}$ and concave in $\mathbf{y}$. For such problems, one can establish strong theoretical guarantees.

In this paper, we focus on a more general case of (1), where $f(\mathbf{x}, \mathbf{y})$ is $\mu$-strongly-concave in $\mathbf{y}$ but possibly nonconvex in $\mathbf{x}$. This case is referred to as stochastic nonconvex-strongly-concave minimax problems, and it is equivalent to the following problem

$$\min_{\mathbf{x} \in \mathbb{R}^d} \left\{ \Phi(\mathbf{x}) \triangleq \max_{\mathbf{y} \in \mathcal{Y}} f(\mathbf{x}, \mathbf{y}) \right\}. \tag{2}$$

Formulation (2) contains several interesting examples in machine learning such as robust optimization [14, 46] and adversarial training [17, 40].

Since $\Phi$ is possibly nonconvex, it is infeasible to find the global minimum in general. One important task of the minimax problem is finding an approximate stationary point of $\Phi$. A simple way to

solve this problem is stochastic gradient descent with max-oracle (SGDmax) [19, 25]. The algorithm includes a nested loop to solve $\max_{\mathbf{y} \in \mathcal{Y}} f(\mathbf{x}, \mathbf{y})$ and use the solution to run approximate stochastic gradient descent (SGD) on $\mathbf{x}$. Lin et al. [25] showed that we can solve problem (2) by directly extending SGD to stochastic gradient descent ascent (SGDA). The iteration of SGDA is just using gradient descent on $\mathbf{x}$ and gradient descent on $\mathbf{y}$. The complexity of SGDA to find $\mathcal{O}(\varepsilon)$-stationary point of $\Phi$ in expectation is $\mathcal{O}\left(\kappa^3 \varepsilon^{-4}\right)$ stochastic gradient evaluations, where $\kappa \triangleq \ell/\mu$ is the condition number. SGDA is more efficient than SGDmax whose complexity is $\mathcal{O}\left((\kappa^3 \varepsilon^{-4}) \log(1/\varepsilon)\right)$.

One insight of SGDA is that the algorithm selects an appropriate ratio of learning rates for $\mathbf{x}$ and $\mathbf{y}$. Concretely, the learning rate for updating $\mathbf{y}$ is $\mathcal{O}(\kappa^2)$ times that of $\mathbf{x}$. Using this idea, it can be shown that the nested loop of SGDmax is unnecessary, and SGDA eliminates the logarithmic term in the complexity result. In addition, Rafique et al. [36] presented some nested-loop algorithms that also achieved $\mathcal{O}\left(\kappa^3 \varepsilon^{-4}\right)$ complexity. Recently, Yan et al. [47] proposed Epoch-GDA which considered constraints on both two variables.

Lin et al. [26] proposed a deterministic algorithm called minimax proximal point algorithm (Minimax PPA) to solve nonconvex-strongly-concave minimax problem whose complexity has square root dependence on $\kappa$. Barazandeh and Razaviyayn [7], Ostrovskii et al. [32], Thekumparampil et al. [42] also studied the non-convex-concave minimax problems, however, these methods do not cover the stochastic setting in this paper and only work for a special case of problem (2) when the stochastic variable $\boldsymbol{\xi}$ is finitely sampled from $\{\boldsymbol{\xi}_1, \ldots, \boldsymbol{\xi}_n\}$ (a.k.a. finite-sum case). That is,

$$f(\mathbf{x}, \mathbf{y}) \triangleq \frac{1}{n} \sum_{i=1}^{n} F(\mathbf{x}, \mathbf{y}; \boldsymbol{\xi}_i). \tag{3}$$

In this paper, we propose a novel algorithm called Stochastic Recursive gradiEnt Descent Ascent (SREDA) for stochastic nonconvex-strongly-concave minimax problems. Unlike SGDmax and SGDA, which only iterate with current stochastic gradients, our SREDA updates the estimator recursively and reduces its variance.

The variance reduction techniques have been widely used in convex and nonconvex minimization problems [1–4, 11, 15, 20, 22, 23, 29, 30, 35, 37, 38, 44, 51, 53] and convex-concave saddle point problems [9, 12, 13, 28, 34]. However, the nonconvex-strongly-concave minimax problems have two variables $\mathbf{x}$ and $\mathbf{y}$ and their roles in the objective function are quite different. To apply the technique of variance reduction, SREDA employs a concave maximizer with multi-step iteration on $\mathbf{y}$ to simultaneously balance the learning rates, gradient batch sizes and iteration numbers of the two variables. We prove SREDA reduces the number of stochastic gradient evaluations to $\mathcal{O}(\kappa^3 \varepsilon^{-3})$, which is the best known upper bound complexity. The result gives optimal dependency on $\varepsilon$ since the lower bound of stochastic first order algorithms for general nonconvex optimization is $\mathcal{O}(\varepsilon^{-3})$ [6]. For finite-sum cases, the gradient cost of SREDA is $\mathcal{O}\left(n \log(\kappa/\varepsilon) + \kappa^2 n^{1/2} \varepsilon^{-2}\right)$ when $n \geq \kappa^2$, and $\mathcal{O}\left((\kappa^2 + \kappa n) \varepsilon^{-2}\right)$ when $n \leq \kappa^2$. This result is sharper than Minimax PPA [26] in the case of $n$ is larger than $\kappa^2$. We summarize the comparison of all algorithms in Table 1.

The paper is organized as follows. In Section 2, we present notations and preliminaries. In Section 3, we review the existing work for stochastic nonconvex-strongly-concave optimization and related techniques. In Section 4, we present the SREDA algorithm and the main theoretical result. In Section 5, we give a brief overview of our convergence analysis. In Section 6, we demonstrate the effectiveness of our methods on robust optimization problem. We conclude this work in Section 7.

## 2   Notation and Preliminaries

We first introduce the notations and preliminaries used in this paper. For a differentiable function $f(\mathbf{x}, \mathbf{y})$, we denote the partial gradient of $f$ with respect to $\mathbf{x}$ and $\mathbf{y}$ at $(\mathbf{x}, \mathbf{y})$ as $\nabla_{\mathbf{x}} f(\mathbf{x}, \mathbf{y})$ and $\nabla_{\mathbf{y}} f(\mathbf{x}, \mathbf{y})$ respectively. We use $\|\cdot\|_2$ to denote the Euclidean norm of vectors. For a finite set $\mathcal{S}$, we denote its cardinality as $|\mathcal{S}|$. We assume that the minimax problem (2) satisfies the following assumptions.

**Assumption 1.** *The function $\Phi(\cdot)$ is lower bounded, i.e., we have $\Phi^* = \inf_{\mathbf{x} \in \mathbb{R}^d} \Phi(\mathbf{x}) > -\infty$.*

**Assumption 2.** *The component function $F$ has an average $\ell$-Lipschitz gradient, i.e., there exists a constant $\ell > 0$ such that $\mathbb{E}\|\nabla F(\mathbf{x}, \mathbf{y}; \boldsymbol{\xi}) - \nabla F(\mathbf{x}', \mathbf{y}'; \boldsymbol{\xi})\|_2^2 \leq \ell^2(\|\mathbf{x} - \mathbf{x}'\|_2^2 + \|\mathbf{y} - \mathbf{y}'\|_2^2)$ for any $(\mathbf{x}, \mathbf{y})$, $(\mathbf{x}', \mathbf{y}')$ and random vector $\boldsymbol{\xi}$*

Table 1: We present the comparison on stochastic gradient complexities of algorithms to solve stochastic problem (2) and finite-sum problem (3). We use notation $\tilde{\mathcal{O}}(\cdot)$ to hide logarithmic factors. Some baseline algorithms solve problem (3) without considering the finite-sum structure and we regard the cost of full gradient evaluation is $\mathcal{O}(n)$.

| Algorithm | Stochastic | Finite-sum | Reference |
|:---:|:---:|:---:|:---:|
| SGDmax (GDmax) | $\tilde{\mathcal{O}}(\kappa^3\varepsilon^{-4})$ | $\tilde{\mathcal{O}}(\kappa^2 n\varepsilon^{-2})$ | [19, 25] |
| PGSMD / PGSVRG | $\mathcal{O}(\kappa^3\varepsilon^{-4})$ | $\mathcal{O}(\kappa^2 n\varepsilon^{-2})$ | [36] |
| MGDA / HiBSA | – | $\mathcal{O}(\kappa^4 n\varepsilon^{-2})$ | [27, 31] |
| Minimax PPA | – | $\tilde{\mathcal{O}}(\kappa^{1/2} n\varepsilon^{-2})$ | [26] |
| SGDA (GDA) | $\mathcal{O}(\kappa^3\varepsilon^{-4})$ | $\mathcal{O}(\kappa^2 n\varepsilon^{-2})$ | [25] |
| SREDA | $\mathcal{O}(\kappa^3\varepsilon^{-3})$ | $\begin{cases} \tilde{\mathcal{O}}\left(n + \kappa^2 n^{1/2}\varepsilon^{-2}\right), & n \geq \kappa^2 \\ \mathcal{O}\left((\kappa^2 + \kappa n)\varepsilon^{-2}\right), & n \leq \kappa^2 \end{cases}$ | this paper |

**Assumption 3.** *The component function $F$ is concave in $\mathbf{y}$. That is, for any $\mathbf{x}$, $\mathbf{y}$, $\mathbf{y}'$ and random vector $\boldsymbol{\xi}$, we have $F(\mathbf{x}, \mathbf{y}; \boldsymbol{\xi}) \leq F(\mathbf{x}, \mathbf{y}'; \boldsymbol{\xi}) + \langle \nabla_{\mathbf{y}} F(\mathbf{x}, \mathbf{y}'; \boldsymbol{\xi}), \mathbf{y} - \mathbf{y}' \rangle$.*

**Assumption 4.** *The function $f(\mathbf{x}, \mathbf{y})$ is $\mu$-strongly-concave in $\mathbf{y}$. That is, there exists a constant $\mu > 0$ such that for any $\mathbf{x}$, $\mathbf{y}$ and $\mathbf{y}'$, we have $f(\mathbf{x}, \mathbf{y}) \leq f(\mathbf{x}, \mathbf{y}') + \langle \nabla_{\mathbf{y}} f(\mathbf{x}, \mathbf{y}'), \mathbf{y} - \mathbf{y}' \rangle - \frac{\mu}{2} \|\mathbf{y} - \mathbf{y}'\|_2^2$.*

**Assumption 5.** *The gradient of each component function $F(\mathbf{x}, \mathbf{y}; \boldsymbol{\xi})$ has bounded variance. That is, there exists a constant $\sigma > 0$ such that $\mathbb{E} \|\nabla F(\mathbf{x}, \mathbf{y}; \boldsymbol{\xi}) - \nabla f(\mathbf{x}, \mathbf{y})\|_2^2 \leq \sigma^2 < \infty$ for any $\mathbf{x}$, $\mathbf{y}$ and random vector $\boldsymbol{\xi}$.*

Under the assumptions of Lipschitz-gradient and strongly-concavity on $f$, we can show that $\Phi(\cdot)$ also has Lipschitz-gradient.

**Lemma 1** ([25, Lemma 4.3]). *Under Assumptions 2 and 4, the function $\Phi(\cdot) = \max_{\mathbf{y} \in \mathcal{Y}} f(\cdot, \mathbf{y})$ has $(\ell + \kappa\ell)$-Lipschitz gradient. Additionally, the function $\mathbf{y}^*(\cdot) = \arg\max_{\mathbf{y}} f(\cdot, \mathbf{y})$ is unique defined and we have $\nabla\Phi(\cdot) = \nabla_{\mathbf{x}} f(\cdot, \mathbf{y}^*(\cdot))$.*

Since $\Phi$ is differentiable, we may define $\varepsilon$-stationary point based on its gradient. The goal of this paper is to establish a stochastic gradient algorithm that output an $\mathcal{O}(\varepsilon)$-stationary point in expectation.

**Definition 1.** *We call $\mathbf{x}$ an $\mathcal{O}(\varepsilon)$-stationary point of $\Phi$ if $\|\nabla\Phi(\mathbf{x})\|_2 \leq \mathcal{O}(\varepsilon)$.*

We also need the notations of projection and gradient mapping to address the constraint on $\mathcal{Y}$.

**Definition 2.** *We define the projection of $\mathbf{y}$ on to convex set $\mathcal{Y}$ by $\Pi_{\mathcal{Y}}(\mathbf{y}) = \arg\min_{\mathbf{z} \in \mathcal{Y}} \|\mathbf{z} - \mathbf{y}\|_2$.*

**Definition 3.** *We define the gradient mapping of $f$ at $(\mathbf{x}', \mathbf{y}')$ with respect to $\mathbf{y}$ as follows*

$$\mathcal{G}_{\lambda, \mathbf{y}}(\mathbf{x}', \mathbf{y}') = \frac{1}{\lambda}\left(\mathbf{y}' - \Pi_{\mathcal{Y}}\left(\mathbf{y}' + \lambda \nabla_{\mathbf{y}} f(\mathbf{x}', \mathbf{y}')\right)\right), \;\; where \; \lambda > 0.$$

## 3 Related Work

In this section, we review recent works for solving stochastic nonconvex-strongly-convex minimax problem (2) and introduce variance reduction techniques in stochastic optimization.

### 3.1 Nonconvex-Strongly-Concave Minimax

We present SGDmax [19, 25] in Algorithm 1. We can realize the max-oracle by stochastic gradient ascent (SGA) with $\mathcal{O}(\kappa^2\varepsilon^{-2}\log(1/\varepsilon))$ stochastic gradient evaluations to achieve sufficient accuracy. Using $S = \mathcal{O}(\kappa\varepsilon^{-2})$ guarantees that the variance of the stochastic gradients is less than $\mathcal{O}(\kappa^{-1}\varepsilon^2)$. It requires $\mathcal{O}(\kappa\varepsilon^{-2})$ iterations with step size $\eta = \mathcal{O}(1/(\kappa\ell))$ to obtain an $\mathcal{O}(\varepsilon)$-stationary point of $\Phi$. The total stochastic gradient evaluation complexity is $\mathcal{O}(\kappa^3\varepsilon^{-4}\log(1/\varepsilon))$. The procedure of SGDA is shown in Algorithm 2.

Since variables $\mathbf{x}$ and $\mathbf{y}$ are not symmetric, we need to select different step sizes for them. In our case, we choose $\eta = \mathcal{O}(1/(\kappa^2\ell))$ and $\lambda = \mathcal{O}(1/\ell)$. This leads to an $\mathcal{O}(\kappa^3\varepsilon^{-4})$ complexity to obtain an $\mathcal{O}(\varepsilon)$-stationary point with $S = \mathcal{O}(\kappa\varepsilon^{-2})$ and $\mathcal{O}(\kappa^2\varepsilon^{-2})$ iterations [25]. Rafique et al. proposed proximally guided stochastic mirror descent and variance reduction (PGSMD / PGSVRG) whose complexity is also $\mathcal{O}(\kappa^3\varepsilon^{-4})$. Both of the above algorithms reveal that the key of solving problem (2) efficiently is to update $\mathbf{y}$ much more frequently than $\mathbf{x}$. The natural intuition is that finding stationary point of a nonconvex function is typically more difficult than finding that of a concave or convex function. SGDmax implements it by updating $\mathbf{y}$ more frequently (SGA in max-oracle) while SGDA iterates $\mathbf{y}$ with a larger step size such that $\lambda/\eta = \mathcal{O}(\kappa^2)$.

## 3.2 Variance Reduction Techniques

Variance reduction techniques has been widely used in stochastic optimization [2, 4, 15, 22, 23, 29, 30, 35, 37]. One scheme of this type of methods is StochAstic Recursive grAdient algoritHm (SARAH) [29, 30]. Nguyen et al. [29] first proposed it for convex minimization and established a convergence result. For nonconvex optimization, a closely related method is Stochastic Path-Integrated Differential EstimatoR (SPIDER) [15]. The algorithm estimates the gradient recursively together with a normalization rule, which guarantees the approximation error of the gradient is $\mathcal{O}(\varepsilon^2)$ at each step. As a result, it can find $\mathcal{O}(\varepsilon)$-stationary point of the nonconvex objective in $\mathcal{O}(\varepsilon^{-3})$ complexity, which matches the lower bound [6]. This idea can also be extended to nonsmooth cases [35, 44].

It is also possible to employ variance reduction to solve minimax problems. Most of the existing works focused on the convex-concave case. For example, Chavdarova et al. [9], Palaniappan and Bach [34], extend SVRG [20, 51] and SAGA [11] to solving strongly-convex-strongly-concave minimax problem in the finite-sum case, and established a linear convergence. One may also use the Catalyst framework [24, 34] and proximal point iteration [10, 28] to further accelerate when the problem is ill-conditioned. Du and Hu [12], Du et al. [13] pointed out that for some special cases, the strongly-convex and strongly-concave assumptions of linear convergence for minimax problem may not be necessary. Additionally, Zhang and Xiao [49] solved multi-level composite optimization problems by variance reduction, but the oracles in their algorithms are different from our settings.

---

**Algorithm 1** SGDmax

---

1: **Input** initial point $\mathbf{x}_0$, learning rate $\eta > 0$, batch size $S > 0$, max-oracle accuracy $\zeta$
2: **for** $k = 0, \ldots, K$ **do**
3:    draw $S$ samples $\{\boldsymbol{\xi}_1, \ldots, \boldsymbol{\xi}_S\}$
4:    find $\mathbf{y}_k$ so that $\mathbb{E}[f(\mathbf{x}_k, \mathbf{y}_k)] \geq \max_{\mathbf{y} \in \mathcal{Y}} f(\mathbf{x}_k, \mathbf{y}) - \zeta$
5:    $\mathbf{x}_{k+1} = \mathbf{x}_k - \eta \cdot \frac{1}{S} \sum_{i=1}^{S} \nabla_{\mathbf{x}} F(\mathbf{x}_k, \mathbf{y}_k; \boldsymbol{\xi}_i)$
6: **end for**
7: **Output** $\hat{\mathbf{x}}$ chosen uniformly at random from $\{\mathbf{x}_i\}_{i=0}^{K}$

---

---

**Algorithm 2** SGDA

---

1: **Input** initial point $(\mathbf{x}_0, \mathbf{y}_0)$, learning rates $\eta > 0$ and $\lambda > 0$, batch size $S > 0$
2: **for** $k = 0, \ldots, K$ **do**
3:    draw $M$ samples $\{\boldsymbol{\xi}_1, \ldots, \boldsymbol{\xi}_S\}$
4:    $\mathbf{x}_{k+1} = \mathbf{x}_k - \eta \cdot \frac{1}{S} \sum_{i=1}^{S} \nabla_{\mathbf{x}} F(\mathbf{x}_k, \mathbf{y}_k; \boldsymbol{\xi}_i)$
5:    $\mathbf{y}_{k+1} = \Pi_{\mathcal{Y}} \left( \mathbf{y}_k + \lambda \cdot \frac{1}{S} \sum_{i=1}^{S} \nabla_{\mathbf{y}} F(\mathbf{x}_k, \mathbf{y}_k; \boldsymbol{\xi}_i) \right)$
6: **end for**
7: **Output** $\hat{\mathbf{x}}$ chosen uniformly at random from $\{\mathbf{x}_i\}_{i=0}^{K}$

---

---

**Algorithm 3** SREDA

---

1: **Input** initial point $\mathbf{x}_0$, learning rates $\eta_k, \lambda > 0$, batch sizes $S_1, S_2 > 0$; periods $q, m > 0$, number of initial iterations $K_0$
2: $\mathbf{y}_0 = \text{PiSARAH}\left(-f(\mathbf{x}_k, \cdot), \ K_0\right)$
3: **for** $k = 0, \ldots, K - 1$ **do**
4:    **if** $\mod (k, q) = 0$
5:       draw $S_1$ samples $\{\boldsymbol{\xi}_1, \ldots, \boldsymbol{\xi}_{S_1}\}$
6:       $\mathbf{v}_k = \frac{1}{S_1} \sum_{i=1}^{S_1} \nabla_{\mathbf{x}} F(\mathbf{x}_k, \mathbf{y}_k; \boldsymbol{\xi}_i)$
7:       $\mathbf{u}_k = \frac{1}{S_1} \sum_{i=1}^{S_1} \nabla_{\mathbf{y}} F(\mathbf{x}_k, \mathbf{y}_k; \boldsymbol{\xi}_i)$
8:    **else**
9:       $\mathbf{v}_k = \mathbf{v}_k'$
10:      $\mathbf{u}_k = \mathbf{u}_k'$
11:   **end if**
12:   $\mathbf{x}_{k+1} = \mathbf{x}_k - \eta_k \mathbf{v}_k$
13:   $(\mathbf{y}_{k+1}, \mathbf{v}_{k+1}', \mathbf{u}_{k+1}') = \text{ConcaveMaximizer}\,(k, m, S_2, \mathbf{x}_k, \mathbf{x}_{k+1}, \mathbf{y}_k, \mathbf{u}_k, \mathbf{v}_k)$
14: **end for**
15: **Output** $\hat{\mathbf{x}}$ chosen uniformly at random from $\{\mathbf{x}_i\}_{i=0}^{K-1}$

---

## 4 Algorithms and Main Results

In this section, we propose a novel algorithm for solving problem (2), which we call Stochastic Recursive gradiEnt Descent Ascent (SREDA). We show that the algorithm finds an $\mathcal{O}(\varepsilon)$-stationary point with a complexity of $\mathcal{O}(\kappa^3 \varepsilon^{-3})$ stochastic gradient evaluations, and this result may be extended to the finite-sum case (3).

### 4.1 Stochastic Recursive Gradient Descent Ascent

SREDA uses variance reduction to track the gradient estimator recursively. Because there are two variables $\mathbf{x}$ and $\mathbf{y}$ in our problem (2), it is not efficient to combine SGDA with SPIDER [15] or (inexact) SARAH [29, 30] directly. The algorithm should approximate the gradient of $f(\mathbf{x}_k, \mathbf{y}_k)$ with small error, and keep the value of $f(\mathbf{x}_k, \mathbf{y}_k)$ sufficiently close to $\Phi(\mathbf{x}_k)$. To achieve this, in the proposed method SREDA, we employ a concave maximizer with stochastic variance reduced gradient ascent on $\mathbf{y}$. The details of SREDA and the concave maximizer are presented in Algorithm 3 and Algorithm 4 respectively. In the rest of this section, we show SREDA can find an $\mathcal{O}(\varepsilon)$-stationary point in $\mathcal{O}(\kappa^3 \varepsilon^{-3})$ stochastic gradient evaluations.

In the initialization of SREDA, we hope to obtain $\mathbf{y}_0 \approx \arg\max_{\mathbf{y} \in \mathcal{Y}} f(\mathbf{x}_0, \mathbf{y}_0)$ for given $\mathbf{x}_0$ such that $\mathbb{E} \|\mathcal{G}_{\lambda, \mathbf{y}}(\mathbf{x}_0, \mathbf{y}_0)\|_2^2 \leq \mathcal{O}(\kappa^{-2} \varepsilon^2)$ . We proposed a new algorithm called projected inexact SARAH (PiSARAH) to address it. PiSARAH extends inexact SARAH (iSARAH) [30] to constrained case, which could achieve the desired accuracy of our initialization with a complexity of $\mathcal{O}(\kappa^2 \varepsilon^{-2} \log(\kappa / \varepsilon))$. We present the details of PiSARAH in Appendix C.

SREDA estimates the gradient of $f(\mathbf{x}_k, \mathbf{y}_k)$ by $(\mathbf{v}_k, \mathbf{u}_k) \approx (\nabla_{\mathbf{x}} f(\mathbf{x}_k, \mathbf{y}_k), \nabla_{\mathbf{y}} f(\mathbf{x}_k, \mathbf{y}_k))$. As illustrated in Algorithm 4, we evaluate the gradient of $f$ with a large batch size $S_1 = \mathcal{O}(\kappa^2 \varepsilon^{-2})$ at the beginning of each period, and update the gradient estimate recursively in concave maximizer with a smaller batch size $S_2 = \mathcal{O}(\kappa \varepsilon^{-1})$.

For variable $\mathbf{x}_k$, we adopt a normalized stochastic gradient descent with a learning rate for theoretical analysis:

$$\eta_k = \min\left(\frac{\varepsilon}{\ell \|\mathbf{v}_k\|_2}, \frac{1}{2\ell}\right) \cdot \mathcal{O}(\kappa^{-1}).$$

With this step size, the change of $\mathbf{x}_k$ is not dramatic at each iteration, which leads to accurate gradient estimates. To simplify implementations of the algorithm, we can also use a fixed learning rate in practical.

---
**Algorithm 4** ConcaveMaximizer $(k, m, S_2, \mathbf{x}_k, \mathbf{x}_{k+1}, \mathbf{y}_k, \mathbf{u}_k, \mathbf{v}_k)$

---
1: **Initialize** $\tilde{\mathbf{x}}_{k,-1} = \mathbf{x}_k, \tilde{\mathbf{y}}_{k,-1} = \mathbf{y}_k, \tilde{\mathbf{x}}_{k,0} = \mathbf{x}_{k+1}, \tilde{\mathbf{y}}_{k,0} = \mathbf{y}_k.$
2: draw $S_2$ samples $\{\boldsymbol{\xi}_1, \ldots, \boldsymbol{\xi}_{S_2}\}$
3: $\tilde{\mathbf{v}}_{k,0} = \mathbf{v}_k + \frac{1}{S_2}\sum_{i=1}^{S_2} \nabla_{\mathbf{x}} F(\tilde{\mathbf{x}}_{k,0}, \tilde{\mathbf{y}}_{k,0}; \boldsymbol{\xi}_i) - \frac{1}{S_2}\sum_{i=1}^{S_2} \nabla_{\mathbf{x}} F(\tilde{\mathbf{x}}_{k,-1}, \tilde{\mathbf{y}}_{k,-1}; \boldsymbol{\xi}_i)$
4: $\tilde{\mathbf{u}}_{k,0} = \mathbf{u}_k + \frac{1}{S_2}\sum_{i=1}^{S_2} \nabla_{\mathbf{y}} F(\tilde{\mathbf{x}}_{k,0}, \tilde{\mathbf{y}}_{k,0}; \boldsymbol{\xi}_i) - \frac{1}{S_2}\sum_{i=1}^{S_2} \nabla_{\mathbf{y}} F(\tilde{\mathbf{x}}_{k,-1}, \tilde{\mathbf{y}}_{k,-1}; \boldsymbol{\xi}_i)$
5: $\tilde{\mathbf{x}}_{k,1} = \tilde{\mathbf{x}}_{k,0}$
6: $\tilde{\mathbf{y}}_{k,1} = \Pi_{\mathcal{Y}}\left(\tilde{\mathbf{y}}_{k,0} + \lambda\tilde{\mathbf{u}}_{k,0}\right)$
7: **for** $t = 1, \ldots, m+1$ **do**
8:     draw $S_2$ samples $\{\boldsymbol{\xi}_{t,1}, \ldots, \boldsymbol{\xi}_{t,S_2}\}$
9:     $\tilde{\mathbf{v}}_{k,t} = \tilde{\mathbf{v}}_{k,t-1} + \frac{1}{S_2}\sum_{i=1}^{S_2} \nabla_{\mathbf{x}} F(\tilde{\mathbf{x}}_{k,t}, \tilde{\mathbf{y}}_{k,t}; \boldsymbol{\xi}_{t,i}) - \frac{1}{S_2}\sum_{i=1}^{S_2} \nabla_{\mathbf{x}} F(\tilde{\mathbf{x}}_{k,t-1}, \tilde{\mathbf{y}}_{k,t-1}; \boldsymbol{\xi}_{t,i})$
10:     $\tilde{\mathbf{u}}_{k,t} = \tilde{\mathbf{u}}_{k,t-1} + \frac{1}{S_2}\sum_{i=1}^{S_2} \nabla_{\mathbf{y}} F(\tilde{\mathbf{x}}_{k,t}, \tilde{\mathbf{y}}_{k,t}; \boldsymbol{\xi}_{t,i}) - \frac{1}{S_2}\sum_{i=1}^{S_2} \nabla_{\mathbf{y}} F(\tilde{\mathbf{x}}_{k,t-1}, \tilde{\mathbf{y}}_{k,t-1}; \boldsymbol{\xi}_{t,i})$
11:     $\tilde{\mathbf{x}}_{k,t+1} = \tilde{\mathbf{x}}_{k,t}$
12:     $\tilde{\mathbf{y}}_{k,t+1} = \Pi_{\mathcal{Y}}\left(\tilde{\mathbf{y}}_{k,t} + \lambda\tilde{\mathbf{u}}_{k,t}\right)$
13: **end for**
14: **Output** $\tilde{\mathbf{y}}_{k,s_k}, \tilde{\mathbf{v}}_{k,s_k}$ and $\tilde{\mathbf{u}}_{k,s_k}$ where $s_k$ is chosen uniformly at random from $\{1, \ldots, m\}$

---

For variable $\mathbf{y}_k$, we additionally expect $f(\mathbf{x}_k, \mathbf{y}_k)$ is a good approximation of $\Phi(\mathbf{x}_k)$, which implies the gradient mapping with respect to $\mathbf{y}_k$ should be small enough. We hope to maintain the inequality $\mathbb{E}\left\|\mathcal{G}_{\lambda,\mathbf{y}}(\mathbf{x}_k, \mathbf{y}_k)\right\|_2^2 \leq \mathcal{O}(\kappa^{-2}\varepsilon^2)$ holds. Hence, we include a multi-step concave maximizer to update $\mathbf{y}$ whose details given in Algorithm 4. This procedure can be regarded as one epoch of PiSARAH. We choose the step size $\lambda = \mathcal{O}(1/\ell)$ for inner iterations, which simultaneously ensure that the gradient mapping with respect to $\mathbf{y}$ is small enough and the change of $\mathbf{y}$ is not dramatic.

### 4.2 Complexity Analysis

As shown in Algorithm 3, SREDA updates variables with a large batch size per $q$ iterations. We choose $q = \mathcal{O}(\varepsilon^{-1})$ as a balance between the number of large batch evaluations with $S_1 = \mathcal{O}(\kappa^2\varepsilon^{-2})$ samples and the concave maximizer with $\mathcal{O}(\kappa)$ iterations and $S_2 = \mathcal{O}(\kappa\varepsilon^{-1})$ samples.

Based on above parameter setting, we can obtain an approximate stationary point $\hat{\mathbf{x}}$ in expectation such that $\mathbb{E}\left\|\nabla\Phi(\hat{\mathbf{x}})\right\|_2 \leq \mathcal{O}(\varepsilon)$ with $K = \mathcal{O}(\kappa\varepsilon^{-2})$ outer iterations. The total number of stochastic gradient evaluations of SREDA comes from the initial run of PiSARAH, large batch gradient evaluation ($S_1$ samples) and concave maximizer. That is,

$$\mathcal{O}(\kappa^2\varepsilon^{-2}\log(\kappa/\varepsilon)) + \mathcal{O}\left(K/q \cdot S_1\right) + \mathcal{O}\left(K \cdot S_2 \cdot m\right) = \mathcal{O}(\kappa^3\varepsilon^{-3}).$$

Let $\Delta_f = f(\mathbf{x}_0, \mathbf{y}_0) + \frac{134\varepsilon^2}{\kappa\ell} - \Phi^*$, then we formally present the main result in Theorem 1.

**Theorem 1.** *Under Assumptions 1-5 with the following parameter choices:*

$$\zeta = \kappa^{-2}\varepsilon^2, \ \eta_k = \min\left(\frac{\varepsilon}{5\kappa\ell\left\|\mathbf{v}_k\right\|_2}, \frac{1}{10\kappa\ell}\right), \ \lambda = \frac{1}{8\ell}, \ S_1 = \left\lceil\frac{2250}{19}\sigma^2\kappa^{-2}\varepsilon^2\right\rceil,$$

$$S_2 = \left\lceil\frac{3687}{76}\kappa q\right\rceil, \ q = \left\lceil\varepsilon^{-1}\right\rceil, \ K = \left\lceil\frac{100\kappa\ell\varepsilon^{-2}\Delta_f}{9}\right\rceil \ and \ m = \lceil 1024\kappa\rceil,$$

*Algorithm 3 outputs $\hat{\mathbf{x}}$ such that $\mathbb{E}\left\|\nabla\Phi(\hat{\mathbf{x}})\right\|_2 \leq 1504\varepsilon$ with $\mathcal{O}(\kappa^3\varepsilon^{-3})$ stochastic gradient evaluations.*

We should point out the complexity shown in Theorem 1 gives optimal dependency on $\varepsilon$. We consider the special case of minimax problem whose objective function has the form

$$f(\mathbf{x}, \mathbf{y}) = g(\mathbf{x}) + h(\mathbf{y})$$

where $g$ is possibly nonconvex and $h$ is strongly-concave, which leads to minimizing on $\mathbf{x}$ and maximizing on $\mathbf{y}$ are independent.

---
**Algorithm 5** SREDA (Finite-sum Case)
---
1: **Input** initial point $\mathbf{x}_0$, learning rates $\eta_k, \lambda > 0$, batch sizes $S_1, S_2 > 0$; periods $q, m > 0$; number of initial iterations $K_0$
2: $\mathbf{y}_0 = \mathrm{PSARAH}\left(-f(\mathbf{x}_k, \cdot),\ K_0\right)$
3: **for** $k = 0, \ldots, K - 1$ **do**
4:    **if** $\mod(k, q) = 0$
5:       $\mathbf{v}_k = \nabla_{\mathbf{x}} f(\mathbf{x}_k, \mathbf{y}_k)$
6:       $\mathbf{u}_k = \nabla_{\mathbf{y}} f(\mathbf{x}_k, \mathbf{y}_k)$
7:    **else**
8:       $\mathbf{v}_k = \mathbf{v}'_k$
9:       $\mathbf{u}_k = \mathbf{u}'_k$
10:   **end if**
11:   $\mathbf{x}_{k+1} = \mathbf{x}_k - \eta_k \mathbf{v}_k$
12:   $(\mathbf{y}_{k+1}, \mathbf{v}'_{k+1}, \mathbf{u}'_{k+1}) = \mathrm{ConcaveMaximizer}\left(k, m, S_2, \mathbf{x}_k, \mathbf{x}_{k+1}, \mathbf{y}_k, \mathbf{u}_k, \mathbf{v}_k\right)$
13: **end for**
14: **Output** $\hat{\mathbf{x}}$ chosen uniformly at random from $\{\mathbf{x}_i\}_{i=0}^{K-1}$
---

Consequently, finding $\mathcal{O}(\varepsilon)$-stationary point of the corresponding $\Phi(\mathbf{x})$ can be reduced to finding $\mathcal{O}(\varepsilon)$-stationary point of nonconvex function $g(\mathbf{x})$, which is based on the stochastic first order-oracle $\nabla_{\mathbf{x}} F(\mathbf{x}, \mathbf{y}; \xi) = \nabla g(\mathbf{x}; \xi)$ (this equality holds for any $\mathbf{y}$ since $\mathbf{x}$ and $\mathbf{y}$ are independent). Hence, the analysis of stochastic nonconvex minimization problem [6] based on $\nabla g(\mathbf{x}; \xi)$ can directly lead to the $\mathcal{O}(\varepsilon^{-3})$ lower bound for our minimax problem. We can prove it by constructing the separate function as $f(\mathbf{x}, \mathbf{y}) = g(\mathbf{x}) + h(\mathbf{y})$ where $g$ is the nonconvex function in Arjevani et al.'s [6] lower bound analysis of stochastic nonconvex minimization, and $h$ is an arbitrary smooth, $\mu$-strongly concave function. It is obvious that the lower bound complexity of finding an $\mathcal{O}(\varepsilon)$-stationary point of $\Phi$ is no smaller than that of finding an $\mathcal{O}(\varepsilon)$-stationary point of $g$, which requires at least $\mathcal{O}(\varepsilon^{-3})$ stochastic gradient evaluations [6].

### 4.3 Extension to Finite-sum Case

SREDA also works for nonconvex-strongly-concave minimax optimization in the finite-sum case (3) with little modification of Algorithm 3. We just need to replace line 5-7 of Algorithm 3 with the full gradients, and use projected SARAH (PSARAH)[1] to initialization. We present the details in Algorithm 5. The algorithm is more efficient than Minimax PPA [26] when $n \geq \kappa^2$. We state the result formally in Theorem 2.

**Theorem 2.** *Suppose Assumption 1-4 hold. In the finite-sum case with $n \geq \kappa^2$, we set the parameters*

$$\zeta = \kappa^{-2}\varepsilon^2,\ \eta_k = \min\left(\frac{\varepsilon}{5\kappa\ell\|\mathbf{v}_k\|_2}, \frac{1}{10\kappa\ell}\right),\ \lambda = \frac{2}{7\ell},\ q = \lceil \kappa^{-1} n^{1/2} \rceil,$$

$$S_2 = \left\lceil \frac{3687}{76}\kappa q \right\rceil,\ K = \left\lceil \frac{100\kappa\ell\varepsilon^{-2}\Delta_f}{9} \right\rceil,\ \text{and}\ m = \lceil 1024\kappa \rceil.$$

*Algorithm 5 outputs $\hat{\mathbf{x}}$ such that $\mathbb{E}\|\nabla\Phi(\hat{\mathbf{x}})\|_2 \leq 1504\varepsilon$ with $\mathcal{O}\left(n\log(\kappa/\varepsilon) + \kappa^2 n^{1/2}\varepsilon^{-2}\right)$ stochastic gradient evaluations.*

*In the case of $n \leq \kappa^2$, we set the parameters*

$$\zeta = \kappa^{-2}\varepsilon^2,\ \eta_k = \min\left(\frac{\varepsilon}{5\kappa\ell\|\mathbf{v}_k\|_2}, \frac{1}{10\kappa\ell}\right),\ \lambda = \frac{1}{8\ell},\ q = 1,$$

$$S_2 = 1,\ K = \left\lceil \frac{100\kappa\ell\varepsilon^{-2}\Delta_f}{9} \right\rceil,\ \text{and}\ m = \lceil 1024\kappa \rceil.$$

*Algorithm 5 outputs $\hat{\mathbf{x}}$ such that $\mathbb{E} \|\nabla\Phi(\hat{\mathbf{x}})\|_2 \leq 1504\varepsilon$ with $\mathcal{O}\left((\kappa^2 + \kappa n)\varepsilon^{-2}\right)$ stochastic gradient evaluations.*

# 5 Sketch of Proofs

We present the briefly overview of the proof of Theorem 1. The details are shown in appendix. Different from Lin et al.'s analysis of SGDA [25] which directly considered the value of $\Phi(\mathbf{x}_k)$ and the distance $\|\mathbf{y}_k - \mathbf{y}^*(\mathbf{x}_k)\|_2$, our proof mainly depends on $f(\mathbf{x}_k, \mathbf{y}_k)$ and its gradient. We split the change of objective functions after one iteration on $(\mathbf{x}_k, \mathbf{y}_k)$ into $A_k$ and $B_k$ as follows

$$f(\mathbf{x}_{k+1}, \mathbf{y}_{k+1}) - f(\mathbf{x}_k, \mathbf{y}_k) = \underbrace{f(\mathbf{x}_{k+1}, \mathbf{y}_k) - f(\mathbf{x}_k, \mathbf{y}_k)}_{A_k} + \underbrace{f(\mathbf{x}_{k+1}, \mathbf{y}_{k+1}) - f(\mathbf{x}_{k+1}, \mathbf{y}_k)}_{B_k}, \quad (4)$$

where $A_k$ provides the decrease of function value $f$ and $B_k$ can characterize the difference between $f(\mathbf{x}_{k+1}, \mathbf{y}_{k+1})$ and $\Phi(\mathbf{x}_{k+1})$. We can show that $\mathbb{E}[A_k] \leq -\mathcal{O}(\kappa^{-1}\varepsilon)$ and $\mathbb{E}[B_k] \leq \mathcal{O}(\kappa^{-1}\varepsilon^2/\ell)$. By taking the average of (4) over $k = 0, \ldots, K$, we obtain

$$\frac{1}{K} \sum_{k=0}^{K-1} \mathbb{E} \|\mathbf{v}_k\|_2 \leq \mathcal{O}(\varepsilon).$$

We can also approximate $\mathbb{E} \|\nabla\Phi(\mathbf{x}_k)\|_2$ by $\mathbb{E} \|\mathbf{v}_k\|_2$ with $\mathcal{O}(\varepsilon)$ estimate error. Then the output $\hat{\mathbf{x}}$ of Algorithm 3 satisfies $\mathbb{E} \|\nabla\Phi(\mathbf{x}_k)\|_2 \leq \mathcal{O}(\varepsilon)$. Based on the discussion in Section 4.2, the number of stochastic gradient evaluation is $\mathcal{O}(\kappa^3\varepsilon^{-3})$. We can also use similar idea to prove Theorem 2.

# 6 Numerical Experiments

We conduct the experiments by using distributionally robust optimization with nonconvex regularized logistic loss [5, 14, 21, 46]. Given dataset $\{(\mathbf{a}_i, b_i)\}_{i=1}^n$ where $\mathbf{a}_i \in \mathbb{R}^d$ is the feature of $i$-th sample and $b_i \in \{1, -1\}$ the corresponding label, the minimax formulation is:

$$\min_{\mathbf{x} \in \mathbb{R}^d} \max_{\mathbf{y} \in \mathcal{Y}} f(\mathbf{x}, \mathbf{y}) \triangleq \frac{1}{n} \sum_{i=1}^n \left( y_i l_i(\mathbf{x}) - V(\mathbf{y}) + g(\mathbf{x}) \right),$$

$l_i(\mathbf{x}) = \log(1 + \exp(-b_i \mathbf{a}_i^\top \mathbf{x}))$, $g$ is the nonconvex regularizer [5]:

$$g(\mathbf{x}) = \lambda_2 \sum_{i=1}^d \frac{\alpha x_i^2}{1 + \alpha x_i^2},$$

$V(\mathbf{y}) = \frac{1}{2}\lambda_1 \|n\mathbf{y} - \mathbf{1}\|_2^2$ and $\mathcal{Y} = \{\mathbf{y} \in \mathbb{R}^n : 0 \leq y_i \leq 1, \sum_{i=1}^n y_i = 1\}$ is a simplex. Following Yan et al. [46], Kohler and Lucchi [21]'s settings, we let $\lambda_1 = 1/n^2$, $\lambda_2 = 10^{-3}$ and $\alpha = 10$ for experiments.

We evaluate compared the performance of SREDA with baseline algorithms GDAmax, GDA, SGDA [25] and Minimax PPA [26] on six real-world data sets "a9a", "w8a", "gisette", "mushrooms", "sido0" and "rcv1", whose details are listed in Table 2. The dataset "sido0" comes from Causality Workbench[2] and the others can be downloaded from LIBSVM repository[3]. Our experiments are conducted on a workstation with Intel Xeon Gold 5120 CPU and 256GB memory. We use MATLAB 2018a to run the code and the operating system is Ubuntu 18.04.4 LTS.

The parameters of the algorithms are chosen as follows: The stepsizes of all algorithms are tuned from $\{10^{-3}, 10^{-2}, 10^{-1}, 1\}$ and we keep the stepsize ratio is $\{10, 10^2, 10^3\}$. For stochastic algorithms SGDA and SREDA, the mini-batch size is set with $\{10, 100, 200\}$. For SREDA, we use the finite-sum version (Algorithm 5 with the first case of Theorem 2) and let $q = m = \lceil n/S_2 \rceil$ heuristically. The initialization of SREDA is based on PSARAH with $K_0 = 5$, $b = 1$ and $m = 20$. For Minimax PPA, we tune the proximal parameter from $\{1, 10, 100\}$ and momentum parameter from $\{0.2, 0.5, 0.7\}$. Each inner loop of Minimax PPA has five times Maximin-AG2 which contains five AGD iterations. The results are shown in Figure 1. It is clear that SREDA converges faster than the baseline algorithms.

| datasets | $n$ | $d$ |
|---|---|---|
| a9a | 32,561 | 123 |
| w8a | 49,749 | 300 |
| gisette | 6,000 | 5,000 |
| mushrooms | 8,124 | 112 |
| sido0 | 12,678 | 4,932 |
| rcv1 | 20,242 | 47,236 |

Table 2: Summary of datasets used in our experiments

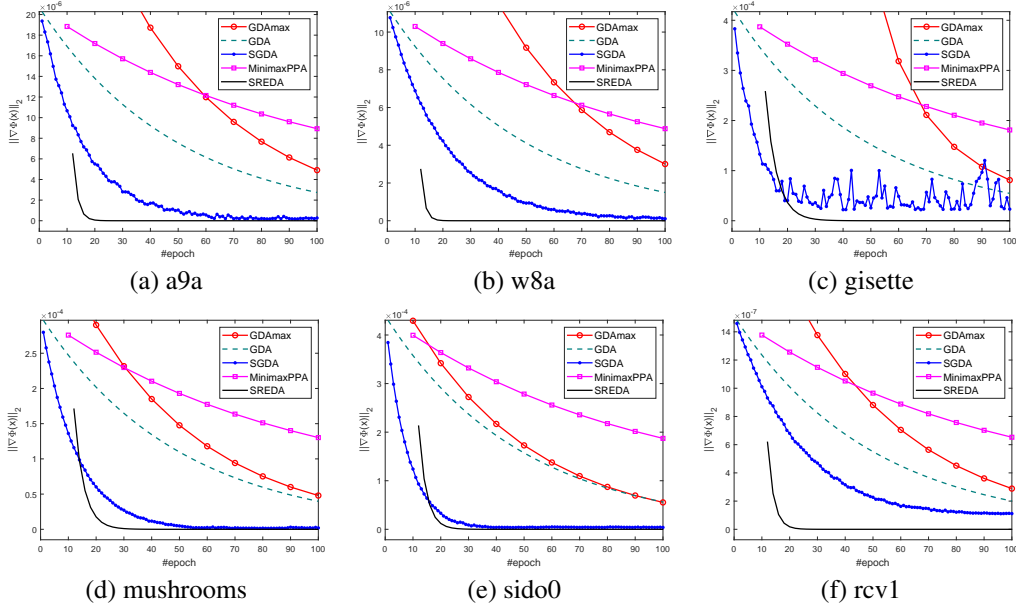

(a) a9a     (b) w8a     (c) gisette

(d) mushrooms     (e) sido0     (f) rcv1

Figure 1: We demonstrate $\|\nabla\Phi(\mathbf{x})\|_2$ vs. the number of epochs for DRO model on real-world datasets "a9a", "w8a", "gisette", "mushrooms", "sido0" and "rcv1" with SREDA and baseline algorithms.

# 7 Conclusion

In this paper, we studied stochastic nonconvex-strongly-concave minimax problems. We proposed a novel algorithm called Stochastic Recursive gradiEnt Descent Ascent (SREDA). The algorithm employs variance reduction to solve minimax problems. Based on the appropriate choice of the parameters, we prove SREDA finds an $\mathcal{O}(\varepsilon)$-stationary point of $\Phi$ with a stochastic gradient complexity of $\mathcal{O}(\kappa^3\varepsilon^{-3})$. This result is better than state-of-the-art algorithms and optimal in its dependency on $\varepsilon$. We can also apply SREDA to the finite-sum case, and show that it performs well when $n$ is larger than $\kappa^2$.

There are still some open problems left. The complexity of SREDA is optimal with respect to $\varepsilon$, but weather it is optimal with respect to $\kappa$ is unknown. It is also possible to employ SREDA to reduce the complexity of stochastic nonconvex-concave minimax problems without the strongly-concave assumption.

## Broader Impact

This paper studied the theory of stochastic minimax optimization. The proposed method SREDA is the first stochastic algorithm which attains the optimal dependency on $\varepsilon$. This observation help us to understand the minimax optimization without convex-concave assumption. It is interesting to apply SREDA to more machine learning applications in future.

## Acknowledgments and Disclosure of Funding

The authors would like to thank Min Tao and Jiahao Xie to point out that the first version of this paper on arXiv has a mistake in the original proof of Theorem 1. This work is supported by GRF 16201320 and the project of Shenzhen Research Institute of Big Data (named "Automated Machine Learning").

## Footnotes

[1]PSARAH extends SARAH [29] to constrained case, which requires $\mathcal{O}\left((n + \kappa)\log(\kappa/\varepsilon)\right)$ stochastic gradient evaluation to achieve sufficient accuracy for our initialization. Please see Appendix E.1 for details

[2]https://www.causality.inf.ethz.ch/challenge.php?page=datasets

[3]https://www.csie.ntu.edu.tw/~cjlin/libsvmtools/datasets/

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
