[Supplementary Material]

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

[4]Note that the proof of Lemma 14 is based on inequality (18) whose left-hand side can be replaced by $\mathbb{E}[f(\mathbf{x}_{k+1}, \mathbf{y}^*) - f(\mathbf{x}_{k+1}, \mathbf{y})]$ for any $\mathbf{y}^* \in \mathcal{Y}$ because of Lemma 9. Hence, we can directly obtain the second inequality of (26) by letting $k = K - 1$ and $\mathbf{y}^* = \mathbf{y}^*(\mathbf{x}_K)$.

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

## Supplementary Materials

This supplementary materials are organized as follows. Appendix A provide several technique lemmas for later analysis. Appendix B give some properties for our concave maximizer. Then, Appendix C proposes projected inexact SARAH (PiSARAH), which generalizes SARAH to constrained optimization and we use it for the initialization of SREDA. Appendix D presents the proof of our main results Theorem 1 and we extend it to prove finite-sum case Theorem 2 in Appendix E.

## A Technical Tools

We first present some useful inequalities in convex optimization, martingale variance bound and gradient mapping.

**Lemma 2** ([29, Theorem 2.1.5 and 2.1.12]). *Suppose $g(\cdot)$ is $\mu$-strongly convex and has $\ell$-Lipschitz gradient. Let $\mathbf{w}^*$ be the minimizer of $g$. Then for any $\mathbf{w}$ and $\mathbf{w}'$, we have the following inequalities*

$$\langle \nabla g(\mathbf{w}) - \nabla g(\mathbf{w}'), \mathbf{w} - \mathbf{w}' \rangle \geq \frac{1}{\ell} \|\nabla g(\mathbf{w}) - \nabla g(\mathbf{w}')\|_2^2, \tag{5}$$

$$\langle \nabla g(\mathbf{w}) - \nabla g(\mathbf{w}'), \mathbf{w} - \mathbf{w}' \rangle \geq \frac{\mu\ell}{\mu+\ell} \|\mathbf{w} - \mathbf{w}'\|_2^2 + \frac{1}{\mu+\ell} \|\nabla g(\mathbf{w}) - \nabla g(\mathbf{w}')\|_2^2, \tag{6}$$

**Lemma 3** ([15, Lemma 1]). *Let $\mathcal{V}_k$ be estimator of $\mathcal{B}(\mathbf{z}_k)$ as*

$$\mathcal{V}_k = \mathcal{B}_{\mathcal{S}_*}(\mathbf{z}_k) - \mathcal{B}_{\mathcal{S}_*}(\mathbf{z}_{k-1}) + \mathcal{V}_{k-1},$$

*where $\mathcal{B}_{\mathcal{S}_*} = \frac{1}{|\mathcal{S}_*|} \sum_{\mathcal{B}_i \in \mathcal{S}_*} \mathcal{B}_i$ satisfies*

$$\mathbb{E}\left[\mathcal{B}_i(\mathbf{z}_k) - \mathcal{B}_i(\mathbf{z}_{k-1}) \mid \mathbf{z}_0, \ldots, \mathbf{z}_{k-1}\right] = \mathbb{E}\left[\mathcal{V}_k - \mathcal{V}_{k-1} \mid \mathbf{z}_0, \ldots, \mathbf{z}_{k-1}\right],$$

*and $\mathcal{B}_i$ is $L$-Lipschitz continuous for any $\mathcal{B}_i \in \mathcal{S}_*$. Then for all $k = 1, \ldots, K$, we have*

$$\mathbb{E}\|\mathcal{V}_k - \mathcal{B}(\mathbf{z}_k) \mid \mathbf{z}_0, \ldots, \mathbf{z}_{k-1}\|_2^2 \leq \|\mathcal{V}_{k-1} - \mathcal{B}(\mathbf{z}_{k-1})\|_2^2 + \frac{\ell^2}{|\mathcal{S}_*|} \mathbb{E}\left[\|\mathbf{z}_k - \mathbf{z}_{k-1}\|_2^2 \mid \mathbf{z}_0, \ldots, \mathbf{z}_{k-1}\right].$$

**Lemma 4** ([29, Corollary 2.2.3]). *Given convex and compact set $\mathcal{C} \subseteq \mathbb{R}^d$ and any $\mathbf{w}, \mathbf{w}' \in \mathbb{R}^d$, we have $\|\Pi_{\mathcal{C}}(\mathbf{w}) - \Pi_{\mathcal{C}}(\mathbf{w}')\|_2 \leq \|\mathbf{w} - \mathbf{w}'\|_2$.*

**Lemma 5** ([29, Corollary 2.2.4]). *Let $g$ has $\ell$-Lipschitz gradient and $\mu$-strongly convex, and $\mathcal{C}$ is a convex set. Denote $\mathcal{G}_\gamma$ be the gradient mapping such that*

$$\mathcal{G}_\gamma(\mathbf{w}) = \frac{\mathbf{w} - \Pi_{\mathcal{C}}(\mathbf{w} - \gamma \nabla g(\mathbf{w}))}{\gamma},$$

*where $\gamma \leq 1/\ell$. Let $\mathbf{w}^* = \arg\min_{\mathbf{w} \in \mathcal{C}} g(\mathbf{w})$, then we have*

$$\langle \mathcal{G}_\gamma(\mathbf{w}), \mathbf{w} - \mathbf{w}^* \rangle^2 \geq \frac{\mu}{2} \|\mathbf{w} - \mathbf{w}^*\|_2 + \frac{1}{2\ell} \|\mathcal{G}_\gamma(\mathbf{w})\|_2^2$$

**Corollary 1.** *Under assumptions of Lemma 5, we have $\frac{\mu}{2} \|\mathbf{w} - \mathbf{w}^*\|_2 \leq \|\mathcal{G}_\gamma(\mathbf{w})\|_2$.*

*Proof.* We have

$$\begin{aligned}
\frac{\mu}{2} \|\mathbf{w} - \mathbf{w}^*\|_2^2 &\leq \langle \mathcal{G}_\gamma(\mathbf{w}), \mathbf{w} - \mathbf{w}^* \rangle - \frac{1}{2\ell} \|\mathcal{G}_\gamma(\mathbf{w})\|_2^2 \\
&\leq \langle \mathcal{G}_\gamma(\mathbf{w}), \mathbf{w} - \mathbf{w}^* \rangle \\
&\leq \|\mathcal{G}_\gamma(\mathbf{w})\|_2 \|\mathbf{w} - \mathbf{w}^*\|_2,
\end{aligned}$$

where the first inequality use Lemma 5 and the last one is based on Cauchy-Schwarz inequality. Then we obtain the desired result. $\square$

# B Some Results of Concave Maximizer

In this section, we present some results of concave maximizer Algorithm 4. The analysis of SREDA is based on the following two auxiliary quantities:

$$\Delta_k = \mathbb{E}\left[\|\mathbf{v}_k - \nabla_{\mathbf{x}} f(\mathbf{x}_k, \mathbf{y}_k)\|_2^2 + \|\mathbf{u}_k - \nabla_{\mathbf{y}} f(\mathbf{x}_k, \mathbf{y}_k)\|_2^2\right] \text{ and } \delta_k = \mathbb{E}\|\mathcal{G}_{\lambda, \mathbf{y}}(\mathbf{x}_k, \mathbf{y}_k)\|_2^2.$$

The main target is to prove both $\Delta_k$ and $\delta_k$ can be bounded by $\mathcal{O}(\kappa^{-2}\varepsilon^2)$.

Using the notations of Algorithm 3 and 4, we denote $\tilde{\mathbf{y}}_k^* = \arg\min_{\mathbf{y} \in \mathcal{Y}} g_k(\mathbf{y})$ and $\hat{\mathbf{u}}_{k,t} = -\tilde{\mathbf{u}}_{k,t}$. It is obvious that $g_k(\cdot)$ is $\mu$-strongly convex and has $\ell$-Lipschitz gradient. The update rule of $\mathbf{x}_k$ in Algorithm 3 means for any $k \geq 0$, we have

$$\|\mathbf{x}_{k+1} - \mathbf{x}_k\|_2^2 \leq \varepsilon_{\mathbf{x}}^2,$$

where $\varepsilon_{\mathbf{x}}^2$ is defined as $\frac{1}{25}\kappa^{-2}\varepsilon^2$. We also denote the gradient mapping with respect to $\mathbf{y}$ as

$$\tilde{\mathcal{G}}_{\lambda, k}(\mathbf{y}) \triangleq \frac{\mathbf{y} - \Pi_{\mathcal{Y}}(\mathbf{y} - \lambda \nabla g_k(\mathbf{y}))}{\lambda} = \mathcal{G}_{\lambda}(\mathbf{x}_{k+1}, \mathbf{y}).$$

We first introduce some lemmas for our iteration and gradient mapping.

**Lemma 6** (Lemma 1 of [23]). *Let* $\mathbf{y}^+ := \Pi_{\mathcal{Y}}(\mathbf{y} - \lambda \mathbf{u})$*, then for all* $\mathbf{z}$*, we have*

$$g_k(\mathbf{y}^+) \leq g_k(\mathbf{z}) + \langle \nabla g_k(\mathbf{y}) - \mathbf{u}, \mathbf{y}^+ - \mathbf{z}\rangle - \frac{\langle \mathbf{y}^+ - \mathbf{y}, \mathbf{y}^+ - \mathbf{z}\rangle}{\lambda} + \frac{\ell}{2}\|\mathbf{y}^+ - \mathbf{y}\|_2^2 + \frac{\ell}{2}\|\mathbf{z} - \mathbf{y}\|_2^2.$$

**Lemma 7** (Lemma 2 of [23]). *Let* $\mathbf{y}^+ := \Pi_{\mathcal{Y}}(\mathbf{y} - \lambda \mathbf{u})$ *and* $\overline{\mathbf{y}_{k,t}} := \Pi_{\mathcal{Y}}(\mathbf{y} - \lambda \nabla g_k(\mathbf{y}))$*, then we have* $\langle \nabla g_k(\mathbf{y}) - \mathbf{u}, \mathbf{y}^+ - \overline{\mathbf{y}_{k,t}}\rangle \leq \lambda \|\nabla g_k(\mathbf{y}) - \mathbf{u}\|_2^2$*.*

**Lemma 8.** *For Algorithm 3 and any* $\mathbf{y}$*, we have* $\|\mathcal{G}_{\lambda, \mathbf{y}}(\mathbf{x}_{k+1}, \mathbf{y}) - \mathcal{G}_{\lambda, \mathbf{y}}(\mathbf{x}_k, \mathbf{y})\|_2^2 \leq \ell^2 \varepsilon_{\mathbf{x}}^2$*.*

*Proof.* Using Lemma 4 and smoothness of $f$, we have

$$\begin{aligned}
&\|\mathcal{G}_{\lambda, \mathbf{y}}(\mathbf{x}_{k+1}, \mathbf{y}_k) - \mathcal{G}_{\lambda, \mathbf{y}}(\mathbf{x}_k, \mathbf{y}_k)\|_2^2 \\
=& \left\|\frac{\mathbf{y}_k - \Pi_{\mathcal{Y}}(\mathbf{y}_k - \lambda \nabla_{\mathbf{y}} f(\mathbf{x}_{k+1}, \mathbf{y}_k))}{\lambda} - \frac{\mathbf{y}_k - \Pi_{\mathcal{Y}}(\mathbf{y}_k - \lambda \nabla_{\mathbf{y}} f(\mathbf{x}_k, \mathbf{y}_k))}{\lambda}\right\|_2^2 \\
\leq& \frac{1}{\lambda^2}\|\Pi_{\mathcal{Y}}(\mathbf{y}_k - \lambda \nabla_{\mathbf{y}} f(\mathbf{x}_{k+1}, \mathbf{y}_k)) - \Pi_{\mathcal{Y}}(\mathbf{y}_k - \lambda \nabla_{\mathbf{y}} f(\mathbf{x}_k, \mathbf{y}_k))\|_2^2 \\
\leq& \|\nabla_{\mathbf{y}} f(\mathbf{x}_{k+1}, \mathbf{y}_k) - \nabla_{\mathbf{y}} f(\mathbf{x}_k, \mathbf{y}_k)\|_2^2 \leq \ell^2 \varepsilon_{\mathbf{x}}^2.
\end{aligned}$$

$\square$

**Lemma 9.** *Let* $\mathbf{y}^+ = \Pi_{\mathcal{Y}}(\mathbf{y} - \lambda \mathbf{u})$ *and* $\lambda < 1/\ell$*, then we have*

$$g_k(\mathbf{y}^+) - g_k(\mathbf{y}^*) \leq \frac{1}{\mu}\left(\|\tilde{\mathcal{G}}_{\lambda, k}(\mathbf{y})\|_2^2 + \|\nabla g_k(\mathbf{y}) - \mathbf{u}\|_2^2\right)$$

*for any* $\mathbf{y}^* \in \mathcal{Y}$*.*

*Proof.* Let $Q(\mathbf{z}) = f(\mathbf{y}) + \langle \mathbf{u}, \mathbf{z} - \mathbf{y}\rangle + \frac{1}{2\lambda}\|\mathbf{z} - \mathbf{y}\|_2^2 + r(\mathbf{z})$. We have $\mathbf{y}^+ = \arg\min_{\mathbf{z}} Q(\mathbf{z})$ and

$$\langle \mathbf{u} + \lambda^{-1}(\mathbf{y}^+ - \mathbf{y}) + \boldsymbol{\xi}, \mathbf{y}^+ - \mathbf{y}^*\rangle = \langle \nabla Q(\mathbf{y}^+), \mathbf{y}^* - \mathbf{y}^+\rangle \geq 0,$$

for any $\mathbf{y}^* \in \mathcal{Y}$. Then

$$\begin{aligned}
&\langle \nabla g_k(\mathbf{y}^+), \mathbf{y}^* - \mathbf{y}^+\rangle \\
=& \langle \nabla g_k(\mathbf{y}^+) - \nabla g_k(\mathbf{y}), \mathbf{y}^* - \mathbf{y}^+\rangle + \langle \nabla g_k(\mathbf{y}), \mathbf{y}^* - \mathbf{y}^+\rangle \\
=& \langle \nabla g_k(\mathbf{y}^+) - \nabla g_k(\mathbf{y}), \mathbf{y}^* - \mathbf{y}^+\rangle + \langle \nabla g_k(\mathbf{y}) - \mathbf{u} + \lambda^{-1}(\mathbf{y} - \mathbf{y}^+), \mathbf{y}^* - \mathbf{y}^+\rangle \\
=& \langle \nabla \tilde{g}_k(\mathbf{y}^+) - \nabla \tilde{g}_k(\mathbf{y}), \mathbf{y}^* - \mathbf{y}^+\rangle + \langle \nabla g_k(\mathbf{y}) - \mathbf{u}, \mathbf{y}^* - \mathbf{y}^+\rangle \\
\geq& -\max(\ell, \lambda^{-1})\|\mathbf{y}^+ - \mathbf{y}\|_2\|\mathbf{y}^* - \mathbf{y}^+\|_2 + \langle \nabla g_k(\mathbf{y}) - \mathbf{u}, \mathbf{y}^* - \mathbf{y}^+\rangle
\end{aligned}$$

$$\geq -\max(\ell, \lambda^{-1}) \left\| \mathbf{y}^+ - \mathbf{y} \right\|_2 \left\| \mathbf{y}^* - \mathbf{y}^+ \right\|_2 - \left\| \nabla g_k(\mathbf{y}) - \mathbf{u} \right\|_2 \left\| \mathbf{y}^* - \mathbf{y}^+ \right\|_2,$$

where $\tilde{g}_k(\mathbf{y}) = g_k(\mathbf{y}) - \frac{1}{2\lambda} \|\mathbf{y}\|_2^2$. The first inequality is due to $\tilde{g}_k$ is at most $\max(\ell, \lambda^{-1})$-smooth, that is

$$-\lambda^{-1}\mathbf{I} \preceq (\mu - \lambda^{-1})\mathbf{I} \preceq \nabla^2 \tilde{g}_k(\mathbf{y}) \preceq (\ell - \lambda^{-1})\mathbf{I} \preceq \ell\mathbf{I}.$$

Consequently, we have

$$
\begin{aligned}
&- \max(\ell, \lambda^{-1}) \left\| \mathbf{y}^+ - \mathbf{y} \right\|_2 \left\| \mathbf{y}^* - \mathbf{y}^+ \right\|_2 \\
\leq &\langle \nabla g_k(\mathbf{y}^+), \mathbf{y}^* - \mathbf{y}^+ \rangle + \left\| \nabla g_k(\mathbf{y}) - \mathbf{u} \right\|_2 \left\| \mathbf{y}^* - \mathbf{y}^+ \right\|_2 \\
\leq &g_k(\mathbf{y}^*) - g_k(\mathbf{y}^+) - \frac{\mu}{2} \left\| \mathbf{y}^* - \mathbf{y}^+ \right\|_2^2 + \left\| \nabla g_k(\mathbf{y}) - \mathbf{u} \right\|_2 \left\| \mathbf{y}^* - \mathbf{y}^+ \right\|_2
\end{aligned}
$$

which implies

$$
\begin{aligned}
& g_k(\mathbf{y}^*) - g_k(\mathbf{y}^+) \\
\geq & \frac{\mu}{2} \left\| \mathbf{y}^* - \mathbf{y}^+ \right\|_2^2 - \max(\ell, \lambda^{-1}) \left\| \mathbf{y}^+ - \mathbf{y} \right\|_2 \left\| \mathbf{y}^* - \mathbf{y}^+ \right\|_2 - \left\| \nabla g_k(\mathbf{y}) - \mathbf{u} \right\|_2 \left\| \mathbf{y}^* - \mathbf{y}^+ \right\|_2 \\
\geq & \inf_{\mathbf{y}} \left\{ \frac{\mu}{2} \left\| \mathbf{z} - \mathbf{y}^+ \right\|_2^2 - \left( \max(\ell, \lambda^{-1}) \left\| \mathbf{y}^+ - \mathbf{y} \right\|_2 + \left\| \nabla g_k(\mathbf{y}) - \mathbf{u} \right\|_2 \right) \left\| \mathbf{z} - \mathbf{y}^+ \right\|_2 \right\} \\
= & -\frac{1}{2\mu} \left( \max(\ell, \lambda^{-1}) \left\| \mathbf{y}^+ - \mathbf{y} \right\|_2 + \left\| \nabla g_k(\mathbf{y}) - \mathbf{u} \right\|_2 \right)^2 \\
\geq & -\frac{1}{\mu} \left( \max(\ell, \lambda^{-1})^2 \left\| \mathbf{y}^+ - \mathbf{y} \right\|_2^2 + \left\| \nabla g_k(\mathbf{y}) - \mathbf{u} \right\|_2^2 \right).
\end{aligned}
$$

Considering that $\eta \leq 1/\ell$, we can obtain the desired result by rearranging above inequality. $\qquad\square$

We can now present the key lemma for the concave maximizer, which upper bounds the magnitude of gradient mapping after one epoch iterations on $\mathbf{y}$.

**Lemma 10.** *For Algorithm 4 with $\lambda = \frac{1}{8\ell}$, we have*

$$
\begin{aligned}
& \mathbb{E}\|\tilde{\mathcal{G}}_{\lambda,k}(\tilde{\mathbf{y}}_{k,s_k})\|_2^2 \\
\leq & \frac{64\ell}{m\mu} \mathbb{E}\|\tilde{\mathcal{G}}_{\lambda,k}(\tilde{\mathbf{y}}_{k,0})\|_2^2 + \left( \frac{64\ell}{m\mu} + 8 \right) \mathbb{E} \left\| \nabla g_k(\tilde{\mathbf{y}}_{k,0}) - \tilde{\mathbf{u}}_{k,0} \right\|_2^2 + 8\ell^2 \mathbb{E} \left\| \tilde{\mathbf{y}}_{k,1} - \tilde{\mathbf{y}}_{k,0} \right\|_2^2.
\end{aligned}
$$

*Proof.* We define $\overline{\mathbf{y}_{k,t}} := \Pi_{\mathcal{Y}}\big(\tilde{\mathbf{y}}_{k,t-1} - \lambda\nabla g_k(\tilde{\mathbf{y}}_{k,t-1})\big)$. The procedure of Algorithm 4 means $\tilde{\mathbf{y}}_{k,t} := \Pi_{\mathcal{Y}}(\tilde{\mathbf{y}}_{k,t-1} - \lambda\hat{\mathbf{u}}_{k,t-1})$. Using Lemma 6 by letting $\mathbf{y}^+ = \tilde{\mathbf{y}}_{k,t}, \mathbf{y} = \tilde{\mathbf{y}}_{k,t-1}, \mathbf{u} = \hat{\mathbf{u}}_{k,t-1}$ and $\mathbf{z} = \overline{\mathbf{y}_{k,t}}$ , we have

$$
\begin{aligned}
g_k(\tilde{\mathbf{y}}_{k,t}) \leq & g_k(\overline{\mathbf{y}_{k,t}}) + \langle \nabla g_k(\tilde{\mathbf{y}}_{k,t-1}) - \tilde{\mathbf{u}}_{k,t-1}, \tilde{\mathbf{y}}_{k,t} - \overline{\mathbf{y}_{k,t}} \rangle \\
& - \frac{1}{\lambda}\langle \tilde{\mathbf{y}}_{k,t} - \tilde{\mathbf{y}}_{k,t-1}, \tilde{\mathbf{y}}_{k,t} - \overline{\mathbf{y}_{k,t}} \rangle + \frac{\ell}{2} \|\tilde{\mathbf{y}}_{k,t} - \tilde{\mathbf{y}}_{k,t-1}\|_2^2 + \frac{\ell}{2} \|\overline{\mathbf{y}_{k,t}} - \tilde{\mathbf{y}}_{k,t-1}\|_2^2.
\end{aligned}
\tag{7}
$$

Using Lemma 6 again by letting $\mathbf{y}^+ = \overline{\mathbf{y}_{k,t}}, \mathbf{y} = \tilde{\mathbf{y}}_{k,t-1}, \mathbf{u} = \nabla g_k(\tilde{\mathbf{y}}_{k,t-1})$ and $\mathbf{z} = \mathbf{y} = \tilde{\mathbf{y}}_{k,t-1}$, we have

$$
\begin{aligned}
g_k(\overline{\mathbf{y}_{k,t}}) \leq & g_k(\tilde{\mathbf{y}}_{k,t-1}) - \frac{1}{\lambda}\langle \overline{\mathbf{y}_{k,t}} - \tilde{\mathbf{y}}_{k,t-1}, \overline{\mathbf{y}_{k,t}} - \tilde{\mathbf{y}}_{k,t-1} \rangle + \frac{\ell}{2} \|\overline{\mathbf{y}_{k,t}} - \tilde{\mathbf{y}}_{k,t-1}\|_2^2 \\
= & g_k(\mathbf{y}_{t-1}) - \left( \frac{1}{\lambda} - \frac{\ell}{2} \right) \|\overline{\mathbf{y}_{k,t}} - \tilde{\mathbf{y}}_{k,t-1}\|_2^2.
\end{aligned}
\tag{8}
$$

Sum over inequalities (7) and (8), we have

$$
\begin{aligned}
& g_k(\tilde{\mathbf{y}}_{k,t}) \\
\leq & g_k(\tilde{\mathbf{y}}_{k,t-1}) + \frac{\ell}{2} \|\tilde{\mathbf{y}}_{k,t} - \tilde{\mathbf{y}}_{k,t-1}\|_2^2 - \left( \frac{1}{\lambda} - \ell \right) \|\overline{\mathbf{y}_{k,t}} - \tilde{\mathbf{y}}_{k,t-1}\|_2^2 \\
& + \langle \nabla g_k(\tilde{\mathbf{y}}_{k,t-1}) - \hat{\mathbf{u}}_{k,t-1}, \tilde{\mathbf{y}}_{k,t} - \overline{\mathbf{y}_{k,t}} \rangle - \frac{1}{\lambda}\langle \tilde{\mathbf{y}}_{k,t} - \tilde{\mathbf{y}}_{k,t-1}, \tilde{\mathbf{y}}_{k,t} - \overline{\mathbf{y}_{k,t}} \rangle^2
\end{aligned}
$$

$$=g_k(\tilde{\mathbf{y}}_{k,t-1}) + \frac{\ell}{2} \|\tilde{\mathbf{y}}_{k,t} - \tilde{\mathbf{y}}_{k,t-1}\|_2^2 - \left(\frac{1}{\lambda} - \ell\right) \|\overline{\mathbf{y}_{k,t}} - \tilde{\mathbf{y}}_{k,t-1}\|_2^2$$

$$+ \langle \nabla g_k(\tilde{\mathbf{y}}_{k,t-1}) - \hat{\mathbf{u}}_{k,t-1}, \tilde{\mathbf{y}}_{k,t} - \overline{\mathbf{y}_{k,t}} \rangle$$

$$- \frac{1}{2\lambda} \left( \|\tilde{\mathbf{y}}_{k,t} - \tilde{\mathbf{y}}_{k,t-1}\|_2^2 + \|\tilde{\mathbf{y}}_{k,t} - \overline{\mathbf{y}_{k,t}}\|_2^2 - \|\overline{\mathbf{y}_{k,t}} - \tilde{\mathbf{y}}_{k,t-1}\|_2^2 \right)$$

$$=g_k(\tilde{\mathbf{y}}_{k,t-1}) - \left(\frac{1}{2\lambda} - \frac{\ell}{2}\right) \|\tilde{\mathbf{y}}_{k,t} - \tilde{\mathbf{y}}_{k,t-1}\|_2^2 - \left(\frac{1}{2\lambda} - \ell\right) \|\overline{\mathbf{y}_{k,t}} - \tilde{\mathbf{y}}_{k,t-1}\|_2^2$$

$$+ \langle \nabla g_k(\tilde{\mathbf{y}}_{k,t-1}) - \hat{\mathbf{u}}_{k,t-1}, \tilde{\mathbf{y}}_{k,t} - \overline{\mathbf{y}_{k,t}} \rangle - \frac{1}{2\lambda} \|\tilde{\mathbf{y}}_{k,t} - \overline{\mathbf{y}_{k,t}}\|_2^2$$

$$\leq g_k(\tilde{\mathbf{y}}_{k,t-1}) - \left(\frac{1}{2\lambda} - \frac{\ell}{2}\right) \|\tilde{\mathbf{y}}_{k,t} - \tilde{\mathbf{y}}_{k,t-1}\|_2^2 - \left(\frac{1}{2\lambda} - \ell\right) \|\overline{\mathbf{y}_{k,t}} - \tilde{\mathbf{y}}_{k,t-1}\|_2^2$$

$$+ \langle \nabla g_k(\tilde{\mathbf{y}}_{k,t-1}) - \hat{\mathbf{u}}_{k,t-1}, \tilde{\mathbf{y}}_{k,t} - \overline{\mathbf{y}_{k,t}} \rangle - \frac{1}{8\lambda} \|\tilde{\mathbf{y}}_{k,t} - \tilde{\mathbf{y}}_{k,t-1}\|_2^2 + \frac{1}{6\lambda} \|\overline{\mathbf{y}_{k,t}} - \tilde{\mathbf{y}}_{k,t-1}\|_2^2$$

$$=g_k(\tilde{\mathbf{y}}_{k,t-1}) - \left(\frac{5}{8\lambda} - \frac{\ell}{2}\right) \|\tilde{\mathbf{y}}_{k,t} - \tilde{\mathbf{y}}_{k,t-1}\|_2^2 - \left(\frac{1}{3\lambda} - \ell\right) \|\overline{\mathbf{y}_{k,t}} - \tilde{\mathbf{y}}_{k,t-1}\|_2^2$$

$$+ \langle \nabla g_k(\tilde{\mathbf{y}}_{k,t-1}) - \hat{\mathbf{u}}_{k,t-1}, \tilde{\mathbf{y}}_{k,t} - \overline{\mathbf{y}_{k,t}} \rangle$$

$$\leq g_k(\tilde{\mathbf{y}}_{k,t-1}) - \left(\frac{5}{8\lambda} - \frac{\ell}{2}\right) \|\tilde{\mathbf{y}}_{k,t} - \tilde{\mathbf{y}}_{k,t-1}\|_2^2 - \left(\frac{1}{3\lambda} - \ell\right) \|\overline{\mathbf{y}_{k,t}} - \tilde{\mathbf{y}}_{k,t-1}\|_2^2$$

$$+ \lambda \|\nabla g_k(\tilde{\mathbf{y}}_{k,t-1}) - \hat{\mathbf{u}}_{k,t-1}\|_2^2,$$

where the second inequality uses Young's inequality as follows

$$\|\tilde{\mathbf{y}}_{k,t} - \tilde{\mathbf{y}}_{k,t-1}\|_2^2 \leq \left(1 + \alpha^{-1}\right) \|\overline{\mathbf{y}_{k,t}} - \tilde{\mathbf{y}}_{k,t-1}\|_2^2 + (1 + \alpha) \|\tilde{\mathbf{y}}_{k,t} - \overline{\mathbf{y}_{k,t}}\|_2^2 \quad \text{with } \alpha = 3,$$

and the last inequality holds due to Lemma 7. Take the expectation on above result, we have

$$\mathbb{E}[g_k(\tilde{\mathbf{y}}_{k,t+1})]$$

$$\leq \mathbb{E}\left[ g_k(\tilde{\mathbf{y}}_{k,t}) - \left(\frac{1}{2\lambda} - \frac{\ell}{2}\right) \|\tilde{\mathbf{y}}_{k,t+1} - \tilde{\mathbf{y}}_{k,t}\|_2^2 - \left(\frac{1}{3\lambda} - \ell\right) \|\overline{\mathbf{y}_{k,t+1}} - \tilde{\mathbf{y}}_{k,t}\|_2^2 \right.$$

$$\left. + \lambda \|\nabla g_k(\tilde{\mathbf{y}}_{k,t}) - \hat{\mathbf{u}}_{k,t}\|_2^2 \right] \qquad (9)$$

$$\leq \mathbb{E}\left[ g_k(\tilde{\mathbf{y}}_{k,t}) - \left(\frac{1}{2\lambda} - \frac{\ell}{2}\right) \|\tilde{\mathbf{y}}_{k,t+1} - \tilde{\mathbf{y}}_{k,t}\|_2^2 - \left(\frac{1}{3\lambda} - \ell\right) \|\overline{\mathbf{y}_{k,t+1}} - \tilde{\mathbf{y}}_{k,t}\|_2^2 \right.$$

$$\left. + \lambda \left( \|\nabla g_k(\tilde{\mathbf{y}}_{k,0}) - \tilde{\mathbf{u}}_{k,0}\|_2^2 + \frac{\ell^2}{S_2} \sum_{i=0}^{t-1} \|\tilde{\mathbf{y}}_{k,i+1} - \tilde{\mathbf{y}}_{k,i}\|_2^2 \right) \right],$$

where the second inequality is based on Lemma 3. Summing over (9) with $t$ from 1 to $m$ and relax the upper bound of $i$ to $m$, we obtain

$$\mathbb{E}[g_k(\tilde{\mathbf{y}}_{k,m})]$$

$$\leq \mathbb{E}[g_k(\tilde{\mathbf{y}}_{k,1})] - \sum_{i=1}^{m} \left(\frac{1}{2\lambda} - \frac{\ell}{2} - \frac{\lambda \ell^2 m}{S_2}\right) \mathbb{E} \|\tilde{\mathbf{y}}_{k,i+1} - \tilde{\mathbf{y}}_{k,i}\|_2^2$$

$$- \left(\frac{1}{3\lambda} - \ell\right) \sum_{i=1}^{m} \mathbb{E} \|\overline{\mathbf{y}_{k,i+1}} - \tilde{\mathbf{y}}_{k,i}\|_2^2 + m\lambda \|\nabla g_k(\tilde{\mathbf{y}}_{k,0}) - \hat{\mathbf{u}}_{k,0}\|_2^2 + \frac{\lambda \ell^2 m}{S_2} \mathbb{E} \|\tilde{\mathbf{y}}_{k,1} - \tilde{\mathbf{y}}_{k,0}\|_2^2$$

Consider that $\lambda = \frac{1}{8\ell}$, we further obtain that

$$\mathbb{E}[g_k(\tilde{\mathbf{y}}_{k,m})]$$

$$\leq \mathbb{E}[g_k(\tilde{\mathbf{y}}_{k,1})] - 3\ell \sum_{i=1}^{m} \mathbb{E} \|\tilde{\mathbf{y}}_{k,i+1} - \tilde{\mathbf{y}}_{k,i}\|_2^2 - \ell\lambda^2 \sum_{i=1}^{m} \mathbb{E}\|\tilde{\mathcal{G}}_{\lambda,k}(\tilde{\mathbf{y}}_{k,i})\|_2^2$$

$$+ m\lambda \left\| \nabla g_k(\tilde{\mathbf{y}}_{k,1}) - \hat{\mathbf{u}}_{k,0} \right\|_2^2 + \frac{\lambda \ell^2 m}{S_2} \mathbb{E} \left\| \tilde{\mathbf{y}}_{k,1} - \tilde{\mathbf{y}}_{k,0} \right\|_2^2$$

$$\leq \mathbb{E}[g_k(\tilde{\mathbf{y}}_{k,1})] - \ell \lambda^2 \sum_{i=1}^{m} \mathbb{E}\|\tilde{\mathcal{G}}_{\lambda,k}(\tilde{\mathbf{y}}_{k,i})\|_2^2 + m\lambda \left\| \nabla g_k(\tilde{\mathbf{y}}_{k,1}) - \hat{\mathbf{u}}_{k,0} \right\|_2^2 + \frac{\lambda \ell^2 m}{S_2} \mathbb{E} \left\| \tilde{\mathbf{y}}_{k,1} - \tilde{\mathbf{y}}_{k,0} \right\|_2^2$$

Let $\tilde{\mathbf{y}}_k^* = \arg\min_{\mathbf{y} \in \mathcal{Y}} g_k(\mathbf{y})$, then the above inequality further implies that

$$\sum_{i=1}^{m} \mathbb{E}\|\tilde{\mathcal{G}}_{\lambda,k}(\tilde{\mathbf{y}}_{k,i})\|_2^2 \leq 64\ell \left( \mathbb{E}[g_k(\tilde{\mathbf{y}}_{k,1}) - g_k(\tilde{\mathbf{y}}_k^*)] \right) + 8m \left\| \nabla g_k(\tilde{\mathbf{y}}_{k,0}) - \hat{\mathbf{u}}_{k,0} \right\|_2^2 .$$

Since $\mathbf{y}_{k+1} = \tilde{\mathbf{y}}_{k,s_k}$ and $s_k$ is sampled from $\{1, \ldots, m\}$, we have

$$\mathbb{E}\|\tilde{\mathcal{G}}_{\lambda,k}(\tilde{\mathbf{y}}_{k,s_k})\|_2^2$$

$$= \frac{1}{m} \sum_{i=1}^{m} \mathbb{E}\|\tilde{\mathcal{G}}_{\lambda,k}(\tilde{\mathbf{y}}_{k,i})\|_2^2$$

$$\leq \frac{64\ell}{m} \left( \mathbb{E}[g_k(\tilde{\mathbf{y}}_{k,1}) - g_k(\tilde{\mathbf{y}}_k^*)] \right) + 8\mathbb{E} \left\| \nabla g_k(\tilde{\mathbf{y}}_{k,0}) - \tilde{\mathbf{u}}_{k,0} \right\|_2^2 + \frac{8\ell^2}{S_2} \mathbb{E} \left\| \tilde{\mathbf{y}}_{k,1} - \tilde{\mathbf{y}}_{k,0} \right\|_2^2$$

$$\leq \frac{64\ell}{m\mu} \left( \mathbb{E}\|\tilde{\mathcal{G}}_{\lambda,k}(\tilde{\mathbf{y}}_{k,0})\|_2^2 + \left\| \nabla g_k(\tilde{\mathbf{y}}_{k,0}) - \tilde{\mathbf{u}}_{k,0} \right\|_2^2 \right) + 8\mathbb{E} \left\| \nabla g_k(\tilde{\mathbf{y}}_{k,0}) - \tilde{\mathbf{u}}_{k,0} \right\|_2^2 + 8\ell^2 \mathbb{E} \left\| \tilde{\mathbf{y}}_{k,1} - \tilde{\mathbf{y}}_{k,0} \right\|_2^2$$

$$= \frac{64\ell}{m\mu} \mathbb{E}\|\tilde{\mathcal{G}}_{\lambda,k}(\tilde{\mathbf{y}}_{k,0})\|_2^2 + \left( \frac{64\ell}{m\mu} + 8 \right) \mathbb{E} \left\| \nabla g_k(\tilde{\mathbf{y}}_{k,0}) - \tilde{\mathbf{u}}_{k,0} \right\|_2^2 + 8\ell^2 \mathbb{E} \left\| \tilde{\mathbf{y}}_{k,1} - \tilde{\mathbf{y}}_{k,0} \right\|_2^2 ,$$

where the first inequality is based on Lemma 9 and the second one is due to assumption $S_2 \geq m$. $\quad\square$

We bound the progress of $\tilde{\mathbf{y}}_{k,t}$ by the following lemmas.

**Lemma 11.** *For Algorithm 4 with $\lambda \leq 2/(\mu + \ell)$, we have*

$$\mathbb{E} \left\| \tilde{\mathbf{y}}_{k,t+1} - \tilde{\mathbf{y}}_{k,t} \right\|_2^2 \leq \left( 1 - \frac{2\lambda\mu\ell}{\mu + \ell} \right) \left\| \tilde{\mathbf{y}}_{k,t} - \tilde{\mathbf{y}}_{k,t-1} \right\|_2^2 \quad \text{for any } t \geq 1.$$

*Proof.* Using the notations of Algorithm 4, we define $g_{k,t}(\mathbf{y}) = -\frac{1}{S_2} \sum_{i=1}^{S_2} \nabla_{\mathbf{y}} F(\mathbf{x}_{k+1}, \mathbf{y}; \boldsymbol{\xi}_{t,i})$. Then, we have

$$\mathbb{E} \left\| \tilde{\mathbf{y}}_{k,t+1} - \tilde{\mathbf{y}}_{k,t} \right\|_2^2$$

$$= \mathbb{E} \left\| \Pi_{\mathcal{Y}}(\tilde{\mathbf{y}}_{k,t} - \lambda \hat{\mathbf{u}}_{k,t}) - \Pi_{\mathcal{Y}}(\tilde{\mathbf{y}}_{k,t-1} - \lambda \hat{\mathbf{u}}_{k,t-1}) \right\|_2^2$$

$$\leq \mathbb{E} \left\| (\tilde{\mathbf{y}}_{k,t} - \lambda \hat{\mathbf{u}}_{k,t}) - (\tilde{\mathbf{y}}_{k,t-1} - \lambda \hat{\mathbf{u}}_{k,t-1}) \right\|_2^2$$

$$= \left\| \tilde{\mathbf{y}}_{k,t} - \tilde{\mathbf{y}}_{k,t-1} \right\|_2^2 - 2\lambda \mathbb{E}\langle \tilde{\mathbf{y}}_{k,t} - \tilde{\mathbf{y}}_{k,t-1}, \hat{\mathbf{u}}_{k,t} - \hat{\mathbf{u}}_{k,t-1} \rangle + \lambda^2 \mathbb{E} \left\| \hat{\mathbf{u}}_{k,t} - \hat{\mathbf{u}}_{k,t-1} \right\|_2^2$$

$$= \left\| \tilde{\mathbf{y}}_{k,t} - \tilde{\mathbf{y}}_{k,t-1} \right\|_2^2 - 2\lambda \mathbb{E}\langle \tilde{\mathbf{y}}_{k,t} - \tilde{\mathbf{y}}_{k,t-1}, \nabla g_{k,t-1}(\tilde{\mathbf{y}}_{k,t}) - \nabla g_{k,t-1}(\tilde{\mathbf{y}}_{k,t-1}) \rangle$$

$$\quad + \lambda^2 \mathbb{E} \left\| \nabla g_{k,t-1}(\tilde{\mathbf{y}}_{k,t}) - \nabla g_{k,t-1}(\tilde{\mathbf{y}}_{k,t-1}) \right\|_2^2$$

$$\leq \left\| \tilde{\mathbf{y}}_{k,t} - \tilde{\mathbf{y}}_{k,t-1} \right\|_2^2 - \frac{2\lambda\mu\ell}{\mu + \ell} \mathbb{E} \left\| \tilde{\mathbf{y}}_{k,t} - \tilde{\mathbf{y}}_{k,t-1} \right\|_2^2$$

$$\quad + \left( \lambda^2 - \frac{2\lambda}{\mu + \ell} \right) \mathbb{E} \left\| \nabla g_{k,t-1}(\tilde{\mathbf{y}}_{k,t}) - \nabla g_{k,t-1}(\tilde{\mathbf{y}}_{k,t-1}) \right\|_2^2$$

$$\leq \left( 1 - \frac{2\lambda\mu\ell}{\mu + \ell} \right) \left\| \tilde{\mathbf{y}}_{k,t} - \tilde{\mathbf{y}}_{k,t-1} \right\|_2^2 ,$$

where the first inequality is based on Lemma 4, the second one comes from inequality (6) of Lemma 2 and the last one is due to $\lambda < 2/(\mu + \ell)$. $\quad\square$

Note that our algorithm estimate $\mathcal{G}_{\lambda,\mathbf{y}}(\mathbf{x}_{k+1}, \tilde{\mathbf{y}}_{k,0})$ by $\frac{1}{\lambda}(\tilde{\mathbf{y}}_{k,0} - \tilde{\mathbf{y}}_{k,1})$, whose norm can be bounded as follows:

**Lemma 12.** *For Algorithm 4, we have*

$$\left\| \frac{\tilde{\mathbf{y}}_{k,0} - \tilde{\mathbf{y}}_{k,1}}{\lambda} \right\|_2^2 \leq 3(\Delta_k + 2\ell^2 \varepsilon_{\mathbf{x}}^2 + \delta_k).$$

*Proof.* The fact $\|\mathbf{a} + \mathbf{b} + \mathbf{c}\|_2^2 \le 3\left(\|\mathbf{a}\|_2^2 + \|\mathbf{b}\|_2^2 + \|\mathbf{c}\|_2^2\right)$ means

$$\left\|\frac{\tilde{\mathbf{y}}_{k,0} - \tilde{\mathbf{y}}_{k,1}}{\lambda}\right\|_2^2$$

$$= \left\|\frac{\tilde{\mathbf{y}}_{k,0} - \tilde{\mathbf{y}}_{k,1}}{\lambda} - \mathcal{G}_{\lambda,\mathbf{y}}(\mathbf{x}_{k+1}, \mathbf{y}_k) + \mathcal{G}_{\lambda,\mathbf{y}}(\mathbf{x}_{k+1}, \mathbf{y}_k) - \mathcal{G}_{\lambda,\mathbf{y}}(\mathbf{x}_k, \mathbf{y}_k) + \mathcal{G}_{\lambda,\mathbf{y}}(\mathbf{x}_k, \mathbf{y}_k)\right\|_2^2$$

$$\le 3 \left\|\frac{\tilde{\mathbf{y}}_{k,0} - \tilde{\mathbf{y}}_{k,1}}{\lambda} - \mathcal{G}_{\lambda,\mathbf{y}}(\mathbf{x}_{k+1}, \mathbf{y}_k)\right\|_2^2 + 3 \|\mathcal{G}_{\lambda,\mathbf{y}}(\mathbf{x}_{k+1}, \mathbf{y}_k) - \mathcal{G}_{\lambda,\mathbf{y}}(\mathbf{x}_k, \mathbf{y}_k)\|_2^2 + 3 \|\mathcal{G}_{\lambda,\mathbf{y}}(\mathbf{x}_k, \mathbf{y}_k)\|_2^2.$$

We use Lemma 4 and Lemma 8 to bound the first and the second term respectively, that is

$$\left\|\frac{\tilde{\mathbf{y}}_{k,0} - \tilde{\mathbf{y}}_{k,1}}{\lambda} - \mathcal{G}_{\lambda,\mathbf{y}}(\mathbf{x}_{k+1}, \mathbf{y}_k)\right\|_2^2$$

$$= \left\|\frac{1}{\lambda}(\mathbf{y}_k - \Pi_{\mathcal{Y}}(\mathbf{y}_k - \lambda\tilde{\mathbf{u}}_{k,0})) - \frac{1}{\lambda}(\mathbf{y}_k - \Pi_{\mathcal{Y}}(\mathbf{y}_k - \lambda\nabla_{\mathbf{y}} f(\mathbf{x}_{k+1}, \mathbf{y}_k)))\right\|_2^2$$

$$= \frac{1}{\lambda^2} \|\Pi_{\mathcal{Y}}(\mathbf{y}_k - \lambda\tilde{\mathbf{u}}_{k,0}) - \Pi_{\mathcal{Y}}(\mathbf{y}_k - \lambda\nabla_{\mathbf{y}} f(\mathbf{x}_{k+1}, \mathbf{y}_k))\|_2^2$$

$$\le \|\tilde{\mathbf{u}}_{k,0} - \nabla_{\mathbf{y}} f(\mathbf{x}_{k+1}, \mathbf{y}_k)\|_2^2 = \widetilde{\Delta}_{k,0},$$

and

$$\|\mathcal{G}_{\lambda,\mathbf{y}}(\mathbf{x}_{k+1}, \mathbf{y}_k) - \mathcal{G}_{\lambda,\mathbf{y}}(\mathbf{x}_k, \mathbf{y}_k)\|_2^2 \le \ell^2 \varepsilon_{\mathbf{x}}^2.$$

The third term is $\|\mathcal{G}_{\lambda,\mathbf{y}}(\mathbf{x}_k, \mathbf{y}_k)\|_2^2 = \delta_k$ because of the definition. Hence, we have

$$\left\|\frac{\tilde{\mathbf{y}}_{k,0} - \tilde{\mathbf{y}}_{k,1}}{\lambda}\right\|_2^2 \le 3(\widetilde{\Delta}_{k,0} + \ell^2 \varepsilon_{\mathbf{x}}^2 + \delta_k)$$

$$\le 3(\Delta_k + \frac{\ell^2 \varepsilon_{\mathbf{x}}^2}{S_2} + \ell^2 \varepsilon_{\mathbf{x}}^2 + \delta_k)$$

$$\le 3(\Delta_k + 2\ell^2 \varepsilon_{\mathbf{x}}^2 + \delta_k).$$

$\square$

Then we can establish the recursive relationship of $\Delta_k$ and $\delta_k$.

**Lemma 13.** *For Algorithm 3 and 4 with $\lambda = \frac{1}{8\ell}$, for any $k = k_0 + 1, k_0 + 2, \ldots, k_0 + q - 1$, we have*

$$\Delta_k \le \Delta_{k_0} + \frac{3}{64 S_2(1 - \alpha)} \sum_{i=k_0}^{k-1} \left(\Delta_i + \delta_i + 2\ell^2 \varepsilon_{\mathbf{x}}^2\right) + \frac{(k - k_0)\ell^2 \varepsilon_{\mathbf{x}}^2}{S_2},$$

$$\delta_{k+1} \le \left(\frac{128\ell}{m\mu} + \frac{3}{8}\right)\delta_k + \left(\frac{64\ell}{m\mu} + \frac{67}{8}\right)\Delta_k + \left(\frac{192\ell}{m\mu} + \frac{35}{4}\right)\ell^2 \varepsilon_{\mathbf{x}}^2,$$

*where*

$$\alpha = 1 - \frac{2\lambda\mu\ell}{\mu + \ell}.$$

*Proof.* We define

$$\widetilde{\Delta}_{k,t} = \mathbb{E}\left(\|\tilde{\mathbf{v}}_{k,t} - \nabla_{\mathbf{x}} f(\tilde{\mathbf{x}}_{k,t}, \tilde{\mathbf{y}}_{k,t})\|_2^2 + \|\tilde{\mathbf{u}}_{k,t} - \nabla_{\mathbf{y}} f(\tilde{\mathbf{x}}_{k,t}, \tilde{\mathbf{y}}_{k,t})\|_2^2\right),$$

then we have

$$\widetilde{\Delta}_{k,0}$$

$$= \mathbb{E}\left(\|\tilde{\mathbf{v}}_{k,0} - \nabla_{\mathbf{x}} f(\tilde{\mathbf{x}}_{k,0}, \tilde{\mathbf{y}}_{k,0})\|_2^2 + \|\tilde{\mathbf{u}}_{k,0} - \nabla_{\mathbf{y}} f(\tilde{\mathbf{x}}_{k,0}, \tilde{\mathbf{y}}_{k,0})\|_2^2\right)$$

$$\le \mathbb{E}\left(\|\mathbf{v}_k - \nabla_{\mathbf{x}} f(\mathbf{x}_k, \mathbf{y}_k)\|_2^2 + \|\mathbf{u}_k - \nabla_{\mathbf{y}} f(\mathbf{x}_k, \mathbf{y}_k)\|_2^2\right) + \frac{\ell^2}{S_2} \mathbb{E}\left(\|\tilde{\mathbf{x}}_{k,0} - \mathbf{x}_k\|_2^2 + \|\tilde{\mathbf{y}}_{k,0} - \mathbf{y}_k\|_2^2\right) \quad (10)$$

$$= \Delta_k + \frac{\ell^2}{S_2} \mathbb{E}\left(\|\mathbf{x}_{k+1} - \mathbf{x}_k\|_2^2 + \|\mathbf{y}_k - \mathbf{y}_k\|_2^2\right)$$

$$\le \Delta_k + \frac{\ell^2 \varepsilon_{\mathbf{x}}^2}{S_2},$$

where the first inequality comes from Lemma 3 by letting $\mathcal{B}(\cdot) = \nabla f(\cdot)$ and $\mathcal{V}_k = (\mathbf{v}_k, \mathbf{u}_k)$.

Now for any $k \geq 1$, we have

$$
\begin{aligned}
\Delta_k =& \mathbb{E}\left( \|\mathbf{v}_k - \nabla_\mathbf{x} f(\mathbf{x}_k, \mathbf{y}_k)\|_2^2 + \|\mathbf{u}_k - \nabla_\mathbf{y} f(\mathbf{x}_k, \mathbf{y}_k)\|_2^2 \right) \\
=& \widetilde{\Delta}_{k-1,s_{k-1}+1} \\
\leq& \mathbb{E}\left( \|\tilde{\mathbf{v}}_{k-1,0} - \nabla_\mathbf{y} f(\tilde{\mathbf{x}}_{k-1,0}, \tilde{\mathbf{y}}_{k-1,0})\|_2^2 + \|\tilde{\mathbf{u}}_{k-1,0} - \nabla_\mathbf{y} f(\tilde{\mathbf{x}}_{k-1,0}, \tilde{\mathbf{y}}_{k-1,0})\|_2^2 \right) \\
& + \frac{\ell^2}{S_2} \sum_{t=0}^{s_{k-1}} \left( \|\tilde{\mathbf{x}}_{k-1,t+1} - \tilde{\mathbf{x}}_{k-1,t}\|_2^2 + \|\tilde{\mathbf{y}}_{k-1,t+1} - \tilde{\mathbf{y}}_{k-1,t}\|_2^2 \right) \\
=& \widetilde{\Delta}_{k-1,0} + \frac{\ell^2}{S_2} \sum_{t=0}^{s_{k-1}} \|\tilde{\mathbf{y}}_{k-1,t+1} - \tilde{\mathbf{y}}_{k-1,t}\|_2^2 \\
\leq& \widetilde{\Delta}_{k-1,0} + \frac{\ell^2}{S_2} \sum_{t=0}^{s_k-1} \alpha^t \|\tilde{\mathbf{y}}_{k-1,1} - \tilde{\mathbf{y}}_{k-1,0}\|_2^2 \\
\leq& \widetilde{\Delta}_{k-1,0} + \frac{\ell^2 \lambda^2}{S_2(1-\alpha)} \left\| \frac{\tilde{\mathbf{y}}_{k-1,1} - \tilde{\mathbf{y}}_{k-1,0}}{\lambda} \right\|_2^2,
\end{aligned}
$$

where the first inequality is due to Lemma 3; the second inequality comes from Lemma 11 and the third one is due to basic property of geometric sequence.

Combining above results and Lemma 12, we have

$$
\begin{aligned}
\Delta_k \leq& \widetilde{\Delta}_{k-1,0} + \frac{\ell^2 \lambda^2}{S_2(1-\alpha)} \left\| \frac{\tilde{\mathbf{y}}_{k-1,1} - \tilde{\mathbf{y}}_{k-1,0}}{\lambda} \right\|_2^2 \\
\leq& \Delta_{k-1} + \frac{\ell^2 \varepsilon_\mathbf{x}^2}{S_2} + \frac{3\ell^2 \lambda^2}{S_2(1-\alpha)} \left( \Delta_{k-1} + \delta_{k-1} + 2\ell^2 \varepsilon_\mathbf{x}^2 \right).
\end{aligned}
$$

Summing over the above inequality from $k_0$ to $k$, we can prove the first part of this theorem as follows

$$
\begin{aligned}
\Delta_k \leq& \Delta_{k-1} + \frac{3\ell^2 \lambda^2}{S_2(1-\alpha)} \left( \Delta_{k-1} + \delta_{k-1} + 2\ell^2 \varepsilon_\mathbf{x}^2 \right) + \frac{\ell^2 \varepsilon_\mathbf{x}^2}{S_2} \\
\leq& \Delta_{k_0} + \frac{3}{64 S_2(1-\alpha)} \sum_{i=0}^{k-1} \left( \Delta_i + \delta_i + 2\ell^2 \varepsilon_\mathbf{x}^2 \right) + \frac{(k-k_0)\ell^2 \varepsilon_\mathbf{x}^2}{S_2}.
\end{aligned}
$$

Recall that $\mathbf{y}_{k+1} = \tilde{\mathbf{y}}_{k,s_k}$ and the definition of $g_k$, we achieve the second part of this lemma:

$$
\begin{aligned}
& \delta_{k+1} \\
=& \mathbb{E} \|\mathcal{G}_{\lambda,\mathbf{y}}(\mathbf{x}_{k+1}, \mathbf{y}_{k+1})\|_2^2 \\
\leq& \frac{64\ell}{m\mu} \mathbb{E} \|\mathcal{G}_{\lambda,\mathbf{y}}(\mathbf{x}_{k+1}, \mathbf{y}_k)\|_2^2 + \left( 8 + \frac{64\ell}{m\mu} \right) \mathbb{E} \|\nabla f(\tilde{\mathbf{x}}_{k,0}, \tilde{\mathbf{y}}_{k,0}) - \tilde{\mathbf{u}}_{k,0}\|_2^2 + 8\ell^2 \mathbb{E} \|\tilde{\mathbf{y}}_{k,1} - \tilde{\mathbf{y}}_{k,0}\|_2^2 \\
\leq& \frac{128\ell}{m\mu} \mathbb{E} \left( \|\mathcal{G}_{\lambda,\mathbf{y}}(\mathbf{x}_{k+1}, \mathbf{y}_k) - \mathcal{G}_{\lambda,\mathbf{y}}(\mathbf{x}_k, \mathbf{y}_k)\|_2^2 + \|\mathcal{G}_{\lambda,\mathbf{y}}(\mathbf{x}_k, \mathbf{y}_k)\|_2^2 \right) \\
& + \left( 8 + \frac{64\ell}{m\mu} \right) \widetilde{\Delta}_{k,0} + \frac{3}{8}(\Delta_k + \delta_k + 2\ell^2 \varepsilon_\mathbf{x}^2) \\
\leq& \frac{128\ell}{m\mu} \left( \ell^2 \varepsilon_\mathbf{x}^2 + \delta_k \right) + \left( 8 + \frac{64\ell}{m\mu} \right) \widetilde{\Delta}_{k,0} + \frac{3}{8}(\Delta_k + \delta_k + 2\ell^2 \varepsilon_\mathbf{x}^2) \\
\leq& \frac{128\ell}{m\mu} \left( \ell^2 \varepsilon_\mathbf{x}^2 + \delta_k \right) + \left( 8 + \frac{64\ell}{m\mu} \right) \left( \Delta_k + \frac{\ell^2 \varepsilon_\mathbf{x}^2}{S_2} \right) + \frac{3}{8}(\Delta_k + \delta_k + 2\ell^2 \varepsilon_\mathbf{x}^2) \\
=& \left( \frac{128\ell}{m\mu} + \frac{3}{8} \right) \delta_k + \left( \frac{64\ell}{m\mu} + \frac{67}{8} \right) \Delta_k + \left( \frac{192\ell}{m\mu} + \frac{35}{4} \right) \ell^2 \varepsilon_\mathbf{x}^2,
\end{aligned}
$$

where the first inequality is according to Lemma 10, the second inequality is based on Young's inequality and Lemma 12, the third inequality is due to Lemma 8 and the other steps are based on definitions. $\qquad\square$

Now we can provide the upper bound of $\Delta_k$ and $\delta_k$.

**Corollary 2.** *For Algorithm 3 and Algorithm 4 with*

$$\eta_k = \min\left(\frac{\varepsilon}{5\kappa\ell\left\|\mathbf{v}_k\right\|_2}, \frac{1}{10\kappa\ell}\right), \lambda = \frac{1}{8\ell}, m = \lceil 1024\kappa \rceil, q = \lceil \varepsilon^{-1} \rceil,$$

$$S_1 = \left\lceil \frac{2250}{19}\sigma^2\kappa^{-2}\varepsilon^2 \right\rceil, S_2 = \left\lceil \frac{3687}{76}\kappa q \right\rceil, \quad and \ \delta_0 \le \kappa^{-2}\varepsilon^2,$$

*Then we have $\Delta_k \le \frac{19}{1125}\kappa^{-2}\varepsilon^2$ and $\delta_k \le \kappa^{-2}\varepsilon^2$ for all $k \ge 0$.*

*Proof.* Firstly, we let $k_0$ be the number of round satisfies $\mathrm{mod}(k_0, q) = 0$ in Algorithm 3 and $\alpha$ be the one defined in Lemma 10, such that

$$\alpha = 1 - \frac{2\lambda\mu\ell}{\mu + \ell} \le 1 - \frac{1}{8\kappa}. \tag{11}$$

The choice of $S_1$ indicates we have

$$\Delta_{k_0} < \frac{19}{2250}\kappa^{-2}\varepsilon^2 \tag{12}$$

for all $k_0$. Then we prove the statement by induction.

**Induction base:** The choice of $S_1$ and Assumption 5 means

$$\Delta_{k_0} \le \frac{19}{2250}\kappa^{-2}\varepsilon^2 < \frac{19}{1125}\kappa^{-2}\varepsilon^2.$$

Combing the assumptions of $\delta_0$, we obtain the induction base.

**Induction step:** For any $k \ge 1$, we suppose $\Delta_{k'} \le \frac{19}{1125}\kappa^{-2}\varepsilon^2$ and $\delta_{k'} \le \kappa^{-2}\varepsilon^2$ holds for all $k' < k$. Let $k_0'$ be the largest integer such that $\mathrm{mod}(k_0', q) = 0$ and $k_0' \le k$. Using Lemma 10, and inequalities (11) and (12), we have

$$
\begin{aligned}
\Delta_k \le & \Delta_{k_0'} + \frac{3}{64S_2(1-\alpha)}\sum_{i=k_0}^{k-1}\left(\Delta_i + \delta_i + 2\ell^2\varepsilon_{\mathbf{x}}^2\right) + \frac{(k - k_0)\ell^2\varepsilon_{\mathbf{x}}^2}{S_2} \\
\le & \Delta_{k_0'} + \frac{3q}{64S_2(1-\alpha)}\left(\frac{19}{1125}\kappa^{-2}\varepsilon^2 + \kappa^{-2}\varepsilon^2 + \frac{2}{25}\kappa^{-2}\varepsilon^2\right) + \frac{q}{25S_2}\kappa^{-2}\varepsilon^2 \\
\le & \frac{19}{1125}\kappa^{-2}\varepsilon^2
\end{aligned}
\tag{13}
$$

and

$$
\begin{aligned}
\delta_k \le & \left(\frac{128\ell}{m\mu} + \frac{3}{8}\right)\delta_{k-1} + \left(\frac{67}{8} + \frac{64\ell}{m\mu}\right)\Delta_{k-1} + \left(\frac{35}{4} + \frac{192\ell}{m\mu}\right)\ell^2\varepsilon_{\mathbf{x}}^2 \\
\le & \left(\frac{1}{8} + \frac{3}{8}\right)\delta_{k-1} + \left(\frac{67}{8} + \frac{1}{16}\right)\Delta_{k-1} + \left(\frac{35}{4} + \frac{3}{16}\right)\cdot\frac{\kappa^{-2}\varepsilon^2}{25} \le \kappa^{-2}\varepsilon^2.
\end{aligned}
\tag{14}
$$

$\square$

## C  Initialization via Projected Inexact SARAH

The initialization of SREDA (line 2 of Algorithm 3) can be regarded as solving a stochastic constrained convex minimization (concave maximization) problem. Hence, we consider the following formulation

$$\min_{\mathbf{w}\in\mathcal{C}} g(\mathbf{w}) \triangleq \mathbb{E}[G(\mathbf{w}; \boldsymbol{\xi})], \tag{15}$$

where $\mathcal{C} \subseteq \mathbb{R}^d$ is a compact and convex set and $\boldsymbol{\xi}$ is a random variable.

We suppose the optimal solution is $\mathbf{w}^* = \arg\min_{\mathbf{w}\in\mathcal{C}} g(\mathbf{w})$, the condition number is $\kappa = \mu/\ell$ and the following assumptions hold.

**Assumption 6.** *The component function $G$ has an average $\ell$-Lipschitz gradient, i.e., there exists a constant $\ell > 0$ such that for any $\mathbf{w}$, $\mathbf{w}'$ and random vector $\boldsymbol{\xi}$, we have*

$$\mathbb{E}\|\nabla G(\mathbf{w};\boldsymbol{\xi}) - \nabla F(\mathbf{w}';\boldsymbol{\xi})\|_2^2 \leq \ell^2 \|\mathbf{w} - \mathbf{w}'\|_2^2.$$

**Assumption 7.** *The component function $G$ is convex. That is, for any $\mathbf{w}$, $\mathbf{w}'$ and random vector $\boldsymbol{\xi}$, we have*

$$G(\mathbf{w};\boldsymbol{\xi}) \geq G(\mathbf{w}';\boldsymbol{\xi}) + \langle \nabla G(\mathbf{w}';\boldsymbol{\xi}), \mathbf{w} - \mathbf{w}' \rangle.$$

**Assumption 8.** *The function $g(\mathbf{w})$ is $\mu$-strongly-convex. That is, there exists a constant $\mu > 0$ such that for any $\mathbf{w}$ and $\mathbf{w}'$, we have*

$$g(\mathbf{w}) \geq g(\mathbf{w}') + \langle \nabla g(\mathbf{w}'), \mathbf{w} - \mathbf{w}' \rangle + \frac{\mu}{2}\|\mathbf{w} - \mathbf{w}'\|_2^2.$$

**Assumption 9.** *The gradient of each component function $G(\mathbf{w};\boldsymbol{\xi})$ has bounded variance. That is, there exists a constant $\sigma > 0$ such that for and $\mathbf{w}$ and random vector $\boldsymbol{\xi}$, we have*

$$\mathbb{E}\|\nabla G(\mathbf{w};\boldsymbol{\xi}) - \nabla g(\mathbf{w})\|_2^2 \leq \sigma^2 < \infty.$$

We propose projected inexact SARAH (PiSARAH) to solve problem (15), whose detailed procedure is presented in Algorithm 6.

---

**Algorithm 6** PiSARAH $(g(\cdot), K_0)$

---

1: **Input** $\mathbf{w}_0 \in \mathcal{C}$, learning rate $\gamma > 0$, inner loop size $m$, batch sizes $b_1$
2: **for** $k = 0, \ldots, K_0 - 1$ **do**
3:      draw $b_1$ samples $\{\boldsymbol{\xi}_1, \cdots, \boldsymbol{\xi}_b\}$
4:      $\tilde{\mathbf{w}}_{k,0} = \mathbf{w}_k$
5:      $\tilde{\mathbf{v}}_{k,0} = \frac{1}{b_1} \sum_{i=1}^{b_1} \nabla G(\tilde{\mathbf{w}}_{k,0}; \boldsymbol{\xi}_i)$
6:      $\tilde{\mathbf{w}}_{k,1} = \Pi_{\mathcal{C}}(\tilde{\mathbf{w}}_{k,0} - \gamma \tilde{\mathbf{v}}_{k,0})$
7:      **for** $t = 1, \ldots, m - 1$ **do**
8:          draw sample $\boldsymbol{\xi}_t$
9:          $\tilde{\mathbf{v}}_{k,t} = \tilde{\mathbf{v}}_{k,t-1} + \nabla G(\tilde{\mathbf{w}}_{k,t}; \boldsymbol{\xi}_t) - \nabla G(\tilde{\mathbf{w}}_{k,t-1}; \boldsymbol{\xi}_t)$
10:         $\tilde{\mathbf{w}}_{k,t+1} = \Pi_{\mathcal{C}}(\tilde{\mathbf{w}}_{k,t} - \gamma \tilde{\mathbf{v}}_{k,t})$
11:      **end for**
12:      $\mathbf{w}_{k+1} = \tilde{\mathbf{w}}_{k,s_k}$, where $s_k$ is uniformly sampled from $\{1, \ldots, m\}$
13: **end for**
14: **Output**: $\mathbf{w}_{K_0}$

---

We are interested in the convergence behavior of the gradient mapping, that is

$$\mathcal{G}_\gamma(\mathbf{w}) \triangleq \frac{\mathbf{w} - \Pi_{\mathcal{C}}(\mathbf{w} - \lambda \nabla g(\mathbf{w}))}{\lambda}.$$

The remain of this section provide the convergence analysis of PiSARAH.

Note that each epoch of PiSARAH can be regarded as using ConcaveMaximizer (Algorithm 4) on $-g(\cdot)$. Hence, we can follow the analysis of Lemma 10 to achieve the result as follows.

**Corollary 3.** *For Algorithm 6 with $\gamma = \frac{1}{8\ell}$, we have*

$$\mathbb{E}\|\mathcal{G}_\gamma(\mathbf{w}_{k+1})\|_2^2 \leq \left(\frac{64\ell}{m\mu} + \frac{1}{4}\right)\mathbb{E}\|\mathcal{G}_\gamma(\mathbf{w}_k)\|_2^2 + \left(\frac{64\ell}{m\mu} + \frac{33}{4}\right)\mathbb{E}\|\nabla g(\tilde{\mathbf{w}}_{k,0}) - \tilde{\mathbf{v}}_{k,0}\|_2^2.$$

*Proof.* Using Lemma 10 in the view of $g_k(\cdot) = g(\cdot)$, we have

$$\begin{aligned}
&\mathbb{E}\|\mathcal{G}_\gamma(\mathbf{w}_{k+1})\|_2^2 \\
\leq &\frac{64\ell}{m\mu}\mathbb{E}\|\mathcal{G}_\gamma(\mathbf{w}_k)\|_2^2 + \left(\frac{64\ell}{m\mu} + 8\right)\mathbb{E}\|\nabla g(\tilde{\mathbf{w}}_{k,0}) - \tilde{\mathbf{v}}_{k,0}\|_2^2 + 8\ell^2\mathbb{E}\|\tilde{\mathbf{w}}_{k,1} - \tilde{\mathbf{w}}_{k,0}\|_2^2.
\end{aligned} \tag{16}$$

The last term of (16) can be bounded by

$$\left\| \frac{\tilde{\mathbf{w}}_{k,0} - \tilde{\mathbf{w}}_{k,1}}{\gamma} \right\|_2^2$$

$$\leq 2 \left\| \frac{\tilde{\mathbf{w}}_{k,0} - \tilde{\mathbf{w}}_{k,1}}{\gamma} - \mathcal{G}_\gamma(\mathbf{w}_{k,0}) \right\|_2^2 + 2 \left\| \mathcal{G}_\gamma(\mathbf{w}_{k,0}) \right\|_2^2$$

$$= 2 \left\| \frac{\tilde{\mathbf{w}}_{k,0} - \Pi_{\mathcal{C}}(\mathbf{w}_{k,0} - \lambda \tilde{\mathbf{v}}_{k,0})}{\gamma} - \frac{\mathbf{w}_{k,0} - \Pi_{\mathcal{C}}(\mathbf{w}_{k,0} - \lambda \nabla g(\mathbf{w}_{k,0}))}{\gamma} \right\|_2^2 + 2 \left\| \mathcal{G}_\gamma(\mathbf{w}_{k,0}) \right\|_2^2$$

$$= 2 \left\| \tilde{\mathbf{v}}_{k,0} - \nabla g(\mathbf{w}_{k,0}) \right\|_2^2 + 2 \left\| \mathcal{G}_\gamma(\mathbf{w}_{k,0}) \right\|_2^2,$$

which implies

$$8\ell^2 \mathbb{E} \left\| \tilde{\mathbf{w}}_{k,1} - \tilde{\mathbf{w}}_{k,0} \right\|_2^2 \leq \frac{1}{4} \mathbb{E} \left[ \left\| \tilde{\mathbf{v}}_{k,0} - \nabla g(\mathbf{w}_{k,0}) \right\|_2^2 + \left\| \mathcal{G}_\gamma(\mathbf{w}_{k,0}) \right\|_2^2 \right]. \tag{17}$$

We finish the proof by combining (16) and (17). $\qquad\square$

Then we provide the main result in this section to show the convergence of the gradient mapping.

**Theorem 3.** *For Algorithm 6 with*

$$K_0 = \left\lceil \frac{\log\left(2\zeta^{-1} \|\mathcal{G}_\gamma(\mathbf{w}_0)\|_2^2\right)}{\log 2} \right\rceil, m = \lceil 256\kappa \rceil, \lambda = \frac{1}{8\ell} \text{ and } b_1 = \left\lceil 34\sigma^2\zeta^{-1} \right\rceil.$$

*Then we have* $\mathbb{E}\|\mathcal{G}_\gamma(\mathbf{w}_{K_0})\|_2^2 \leq \zeta.$

*Proof.* Using Corollary 3 with $m = \lceil 256\kappa \rceil$ and $b_1 = \left\lceil 34\sigma^2\zeta^{-1} \right\rceil$ we have

$$\mathbb{E}\|\mathcal{G}_\gamma(\mathbf{w}_{k+1})\|_2^2 \leq \frac{1}{2}\mathbb{E}\|\mathcal{G}_\gamma(\mathbf{w}_k)\|_2^2 + \frac{\zeta}{4},$$

which implies

$$\mathbb{E}\|\mathcal{G}_\gamma(\mathbf{w}_{K_0})\|_2^2 - \frac{\zeta}{2}$$

$$\leq \frac{1}{2}\left( \mathbb{E}\|\mathcal{G}_\gamma(\mathbf{w}_{K_0-1})\|_2^2 - \frac{\zeta}{2} \right)$$

$$\leq \frac{1}{2^{K_0}}\left( \mathbb{E}\|\mathcal{G}_\gamma(\mathbf{w}_0)\|_2^2 - \frac{\zeta}{2} \right)$$

$$\leq \frac{1}{2^{K_0}}\mathbb{E}\|\mathcal{G}_\gamma(\mathbf{w}_0)\|_2^2 \leq \frac{\zeta}{2}.$$

Hence, we have $\mathbb{E}\|\mathcal{G}_\gamma(\mathbf{w}_{K_0})\|_2^2 \leq \zeta.$ $\qquad\square$

The result of Corollary 3 indicate that we hope the PiSARAH as initialization to make the gradient mapping is no larger than $\mathcal{O}(\kappa^{-2}\varepsilon^2)$. The following statement shows we can implement it within $\mathcal{O}(\kappa^2\varepsilon^{-2}\log(\kappa/\varepsilon))$ stochastic gradient evaluations.

**Corollary 4.** *Under assumptions of Theorem 3, we can obtain* $\mathbb{E}\|\mathcal{G}_\gamma(\mathbf{w}_{K_0})\|_2^2 \leq \kappa^{-2}\varepsilon^2$ *with* $\mathcal{O}(\kappa^2\varepsilon^{-2}\log(\kappa/\varepsilon))$ *stochastic gradient evaluations.*

*Proof.* Using Theorem 3 with $\zeta = \kappa^{-2}\varepsilon^2$, we have $\mathbb{E}\|\mathcal{G}_\gamma(\mathbf{w}_{K_0})\|_2^2 \leq \kappa^{-2}\varepsilon^2$. The total number of stochastic gradient evaluation is

$$K_0 \cdot (b_1 + m)$$

$$= \left\lceil \frac{\log\left(2\kappa^2\varepsilon^{-2}\|\mathcal{G}_\gamma(\mathbf{w}_0)\|_2^2\right)}{\log 2} \right\rceil \cdot \left( \left\lceil 34\sigma^2\kappa^2\varepsilon^{-2} \right\rceil + \lceil 256\kappa \rceil \right)$$

$$= \mathcal{O}(\kappa^2\varepsilon^{-2}\log(\kappa/\varepsilon))$$

$\qquad\square$

# D The Proof of Theorem 1

Our proof mainly depends on $f(\mathbf{x}_k, \mathbf{y}_k)$ and its gradient mapping with respect to $\mathbf{y}$, which is different from Lin et al.'s [25] analysis that directly considered the value of $\Phi(\mathbf{x}_k)$ and the distance $\|\mathbf{y}_k - \mathbf{y}^*(\mathbf{x}_k)\|_2$. We split the change of objective functions after one iteration on $(\mathbf{x}_k, \mathbf{y}_k)$ into $A_k$ and $B_k$ as follows

$$f(\mathbf{x}_{k+1}, \mathbf{y}_{k+1}) - f(\mathbf{x}_k, \mathbf{y}_k) = \underbrace{f(\mathbf{x}_{k+1}, \mathbf{y}_k) - f(\mathbf{x}_k, \mathbf{y}_k)}_{A_k} + \underbrace{f(\mathbf{x}_{k+1}, \mathbf{y}_{k+1}) - f(\mathbf{x}_{k+1}, \mathbf{y}_k)}_{B_k},$$

where $A_k$ provides the decrease of function value $f$ and $B_k$ can characterize the difference between $f(\mathbf{x}_{k+1}, \mathbf{y}_{k+1})$ and $\Phi(\mathbf{x}_{k+1})$. We want to show $\mathbb{E}[A_k] \leq -\mathcal{O}\left(\kappa^{-1}\varepsilon\right)$ and $\mathbb{E}[B_k] \leq \mathcal{O}\left((\kappa\ell)^{-1}\varepsilon^2\right)$. Connecting the upper bound of $A_k$ and $B_k$, we can bound the average of $\mathbb{E}\|\mathbf{v}_k\|_2^2$ and use it to prove the upper bound of $\mathbb{E}[\nabla\Phi(\hat{\mathbf{x}})]$ we desired.

We provide two lemmas for preparing the proof of our main results, Theorem 1. The first lemma is to upper bound $B_k$.

**Lemma 14.** *Under assumptions of Theorem 1, we have $\mathbb{E}[B_k] \leq \dfrac{134\varepsilon^2}{\kappa\ell}$ for any $k \geq 1$.*

*Proof.* Note that our algorithm means $\mathbf{y}_k = \tilde{\mathbf{y}}_{k-1, s_{k-1}}$, where $s_{k-1}$ is sampled from $\{1, \ldots, m\}$. Using Lemma 8 by letting $\mathbf{y}^+ = \mathbf{y}_k = \tilde{\mathbf{y}}_{k-1, s_{k-1}}$, $\mathbf{y} = \tilde{\mathbf{y}}_{k-1, s_{k-1}-1}$ and $\mathbf{u} = \tilde{\mathbf{u}}_{k-1, s_{k-1}-1} = -\hat{\mathbf{u}}_{k-1, s_{k-1}-1}$, then we have

$$
\begin{aligned}
&\mathbb{E}[f(\mathbf{x}_{k+1}, \mathbf{y}_{k+1}) - f(\mathbf{x}_{k+1}, \mathbf{y}_k)] \\
=&\mathbb{E}[f(\mathbf{x}_{k+1}, \mathbf{y}_{k+1}) - f(\mathbf{x}_{k+1}, \tilde{\mathbf{y}}_{k-1, s_{k-1}})] \\
\leq&\frac{1}{\mu}\mathbb{E}\left[\left\|\mathcal{G}_{\lambda, \mathbf{y}}(\mathbf{x}_{k+1}, \tilde{\mathbf{y}}_{k-1, s_{k-1}-1})\right\|_2^2 + \left\|\nabla_{\mathbf{y}} f(\mathbf{x}_{k+1}, \tilde{\mathbf{y}}_{k-1, s_{k-1}-1}) - \tilde{\mathbf{u}}_{k-1, s_{k-1}-1}\right\|_2^2\right].
\end{aligned}
\quad (18)
$$

We first bound the first term of (18):

$$
\begin{aligned}
&\mathbb{E}\left\|\mathcal{G}_{\lambda, \mathbf{y}}(\mathbf{x}_{k+1}, \tilde{\mathbf{y}}_{k-1, s_{k-1}-1})\right\|_2^2 \\
\leq&2\mathbb{E}\left\|\mathcal{G}_{\lambda, \mathbf{y}}(\mathbf{x}_{k+1}, \mathbf{y}_k)\right\|_2^2 + 2\mathbb{E}\left\|\mathcal{G}_{\lambda, \mathbf{y}}(\mathbf{x}_{k+1}, \tilde{\mathbf{y}}_{k-1, s_{k-1}-1}) - \mathcal{G}_{\lambda, \mathbf{y}}(\mathbf{x}_{k+1}, \tilde{\mathbf{y}}_{k-1, s_{k-1}})\right\|_2^2 \\
=&2\mathbb{E}\left[\left\|\mathcal{G}_{\lambda}(\mathbf{x}_{k+1}, \mathbf{y}_k) - \mathcal{G}_{\lambda}(\mathbf{x}_k, \mathbf{y}_k)\right\|_2 + \left\|\mathcal{G}_{\lambda}(\mathbf{x}_k, \mathbf{y}_k)\right\|_2^2\right] \\
&+ 2\mathbb{E}\left\|\mathcal{G}_{\lambda}(\mathbf{x}_{k+1}, \tilde{\mathbf{y}}_{k-1, s_{k-1}-1}) - \mathcal{G}_{\lambda}(\mathbf{x}_{k+1}, \tilde{\mathbf{y}}_{k-1, s_{k-1}})\right\|_2^2 \\
\leq&2(\ell^2\varepsilon_{\mathbf{x}}^2 + \delta_k) + 2\mathbb{E}\left\|\mathcal{G}_{\lambda}(\mathbf{x}_{k+1}, \tilde{\mathbf{y}}_{k-1, s_{k-1}-1}) - \mathcal{G}_{\lambda}(\mathbf{x}_{k+1}, \tilde{\mathbf{y}}_{k-1, s_{k-1}})\right\|_2^2,
\end{aligned}
$$

where the inequalities are based on Young's inequality and Lemma 8.

Using similar ideas, we can also prove

$$
\begin{aligned}
&\mathbb{E}\left\|\mathcal{G}_{\lambda, \mathbf{y}}(\mathbf{x}_{k+1}, \tilde{\mathbf{y}}_{k-1, s_{k-1}-1}) - \mathcal{G}_{\lambda, \mathbf{y}}(\mathbf{x}_{k+1}, \tilde{\mathbf{y}}_{k-1, s_{k-1}})\right\|_2^2 \\
\leq&3\mathbb{E}\left\|\mathcal{G}_{\lambda, \mathbf{y}}(\mathbf{x}_{k+1}, \tilde{\mathbf{y}}_{k-1, s_{k-1}-1}) - \mathcal{G}_{\lambda, \mathbf{y}}(\mathbf{x}_k, \tilde{\mathbf{y}}_{k-1, s_{k-1}-1})\right\|_2^2 \\
&+ 3\mathbb{E}\left\|\mathcal{G}_{\lambda, \mathbf{y}}(\mathbf{x}_{k+1}, \tilde{\mathbf{y}}_{k-1, s_{k-1}}) - \mathcal{G}_{\lambda, \mathbf{y}}(\mathbf{x}_k, \tilde{\mathbf{y}}_{k-1, s_{k-1}})\right\|_2^2 \\
&+ 3\mathbb{E}\left\|\mathcal{G}_{\lambda, \mathbf{y}}(\mathbf{x}_k, \tilde{\mathbf{y}}_{k-1, s_{k-1}-1}) - \mathcal{G}_{\lambda, \mathbf{y}}(\mathbf{x}_k, \tilde{\mathbf{y}}_{k-1, s_{k-1}})\right\|_2^2 \\
\leq&6(\ell^2\varepsilon_{\mathbf{x}}^2 + \delta_k) + 3\mathbb{E}\left\|\mathcal{G}_{\lambda, \mathbf{y}}(\mathbf{x}_k, \tilde{\mathbf{y}}_{k-1, s_{k-1}-1}) - \mathcal{G}_{\lambda, \mathbf{y}}(\mathbf{x}_k, \tilde{\mathbf{y}}_{k-1, s_{k-1}})\right\|_2^2,
\end{aligned}
$$

and

$$
\begin{aligned}
&\mathbb{E}\left\|\mathcal{G}_{\lambda, \mathbf{y}}(\mathbf{x}_k, \tilde{\mathbf{y}}_{k-1, s_{k-1}-1}) - \mathcal{G}_{\lambda, \mathbf{y}}(\mathbf{x}_k, \tilde{\mathbf{y}}_{k-1, s_{k-1}})\right\|_2^2 \\
=&\mathbb{E}\left\|\frac{\tilde{\mathbf{y}}_{k-1, s_{k-1}-1} - \Pi_{\mathcal{Y}}(\tilde{\mathbf{y}}_{k-1, s_{k-1}-1} - \lambda\nabla g_k(\tilde{\mathbf{y}}_{k-1, s_{k-1}-1}))}{\lambda}\right. \\
&\left. - \frac{\tilde{\mathbf{y}}_{k-1, s_{k-1}} - \Pi_{\mathcal{Y}}(\tilde{\mathbf{y}}_{k-1, s_{k-1}} - \lambda\nabla g_k(\tilde{\mathbf{y}}_{k-1, s_{k-1}}))}{\lambda}\right\|_2^2
\end{aligned}
$$

$$
\leq 2\mathbb{E}\left\|\frac{\Pi_{\mathcal{Y}}(\tilde{\mathbf{y}}_{k-1,s_{k-1}-1} - \lambda\nabla g_k(\tilde{\mathbf{y}}_{k-1,s_{k-1}-1}))}{\lambda} - \frac{\Pi_{\mathcal{Y}}(\tilde{\mathbf{y}}_{k-1,s_{k-1}} - \lambda\nabla g_k(\tilde{\mathbf{y}}_{k-1,s_{k-1}}))}{\lambda}\right\|_2^2
$$

$$
+ 2\mathbb{E}\left\|\frac{\tilde{\mathbf{y}}_{k-1,s_{k-1}-1} - \tilde{\mathbf{y}}_{k-1,s_{k-1}}}{\lambda}\right\|_2^2
$$

$$
\leq 2\mathbb{E}\left\|\frac{(\tilde{\mathbf{y}}_{k-1,s_{k-1}-1} - \lambda\nabla g_k(\tilde{\mathbf{y}}_{k-1,s_{k-1}-1}))}{\lambda} - \frac{(\tilde{\mathbf{y}}_{k-1,s_{k-1}} - \lambda\nabla g_k(\tilde{\mathbf{y}}_{k-1,s_{k-1}}))}{\lambda}\right\|_2^2
$$

$$
+ 2\mathbb{E}\left\|\frac{\tilde{\mathbf{y}}_{k-1,s_{k-1}-1} - \tilde{\mathbf{y}}_{k-1,s_{k-1}}}{\lambda}\right\|_2^2
$$

$$
\leq 6\mathbb{E}\left\|\frac{\tilde{\mathbf{y}}_{k-1,s_{k-1}-1} - \tilde{\mathbf{y}}_{k-1,s_{k-1}}}{\lambda}\right\|_2^2 + 2\mathbb{E}\left\|\nabla g_k(\tilde{\mathbf{y}}_{k-1,s_{k-1}-1})) - \nabla g_k(\tilde{\mathbf{y}}_{k-1,s_{k-1}}))\right\|_2^2
$$

$$
\leq (6 + 2\ell^2\lambda^2)\mathbb{E}\left\|\frac{\tilde{\mathbf{y}}_{k-1,s_{k-1}-1} - \tilde{\mathbf{y}}_{k-1,s_{k-1}}}{\lambda}\right\|_2^2
$$

$$
\leq (6 + 2\ell^2\lambda^2)\mathbb{E}\left\|\frac{\tilde{\mathbf{y}}_{k-1,1} - \tilde{\mathbf{y}}_{k-1,0}}{\lambda}\right\|_2^2.
$$

Combining all above results, we have

$$
\mathbb{E}\left\|\mathcal{G}_{\lambda,\mathbf{y}}(\mathbf{x}_{k+1}, \tilde{\mathbf{y}}_{k-1,s_{k-1}-1})\right\|_2^2
$$

$$
\leq 14(\ell^2\varepsilon_{\mathbf{x}}^2 + \delta_k) + 6(6 + 2\ell^2\lambda^2)\mathbb{E}\left\|\frac{\tilde{\mathbf{y}}_{k-1,1} - \tilde{\mathbf{y}}_{k-1,0}}{\lambda}\right\|_2^2
$$

$$
\leq 14(\ell^2\varepsilon_{\mathbf{x}}^2 + \delta_k) + 18(6 + 2\ell^2\lambda^2)(\Delta_k + 2\ell^2\varepsilon_{\mathbf{x}}^2 + \delta_k)
$$

$$
\leq 14\left(\frac{1}{25}\kappa^{-2}\varepsilon^2 + \kappa^{-2}\varepsilon^2\right) + \frac{1737}{16}\left(\frac{19}{1125}\kappa^{-2}\varepsilon^2 + \frac{2}{25}\kappa^{-2}\varepsilon^2 + \kappa^{-2}\varepsilon^2\right) \tag{19}
$$

$$
\leq \left(\frac{364}{25} + \frac{1737}{16}\cdot\frac{1234}{1125}\right)\kappa^{-2}\varepsilon^2
$$

$$
= \frac{133641}{1000}\kappa^{-2}\varepsilon^2
$$

where the second inequality is based on Lemma 12 and the third inequality is due to Corollary 3.

We bound the second term of (18) as follows:

$$
\mathbb{E}\left\|\nabla_{\mathbf{y}}f(\mathbf{x}_k, \tilde{\mathbf{y}}_{k-1,s_{k-1}-1}) - \tilde{\mathbf{u}}_{k-1,s_{k-1}-1}\right\|_2^2
$$

$$
\leq \mathbb{E}\left\|\nabla_{\mathbf{y}}f(\mathbf{x}_k, \tilde{\mathbf{y}}_{k-1,s_{k-1}}) - \tilde{\mathbf{u}}_{k-1,s_{k-1}}\right\|_2^2
$$

$$
= \mathbb{E}\left\|\nabla_{\mathbf{y}}f(\mathbf{x}_k, \mathbf{y}_k) - \mathbf{u}_k\right\|_2^2 \tag{20}
$$

$$
\leq \Delta_k \leq \frac{19}{1125}\kappa^{-2}\varepsilon^2,
$$

where the first inequality is based on Lemma 3 and the last one is due to Corollary 3.

By connecting inequalities (18), (19) and (20), we have

$$
\mathbb{E}[B_k] \leq \frac{1}{\mu}\cdot\frac{1202921}{9000}\kappa^{-2}\varepsilon^2 \leq \frac{134\varepsilon^2}{\kappa\ell}.
$$

$\square$

Then we show the estimate error of approximating $\nabla\Phi(\mathbf{x}_k)$ by $\mathbf{v}_k$.

**Lemma 15.** *Under assumptions of Theorem 1, we have*

$$
\mathbb{E}\left\|\nabla\Phi(\mathbf{x}_k)\right\|_2 \leq \mathbb{E}\left\|\mathbf{v}_k\right\|_2 + \frac{15}{7}\varepsilon.
$$

*Proof.* Consider that we have defined $\mathbf{y}^*(\mathbf{x}) = \arg\max_{\mathbf{y} \in \mathcal{Y}} f(\mathbf{x}, \mathbf{y})$, then we have

$$
\begin{aligned}
&\mathbb{E}\left\|\nabla\Phi(\mathbf{x}_k) - \nabla_{\mathbf{x}} f(\mathbf{x}_k, \mathbf{y}_k)\right\|_2^2 \\
=&\mathbb{E}\left\|\nabla_{\mathbf{x}} f(\mathbf{x}_k, \mathbf{y}^*(\mathbf{x}_k)) - \nabla_{\mathbf{x}} f(\mathbf{x}_k, \mathbf{y}_k)\right\|_2^2 \\
\leq&\ell^2 \mathbb{E}\left\|\mathbf{y}^*(\mathbf{x}_k) - \mathbf{y}_k\right\|_2^2 \\
\leq&\frac{4\ell^2}{\mu^2}\mathbb{E}\left\|\mathcal{G}_{\lambda,\mathbf{y}}(\mathbf{x}_k, \mathbf{y}_k)\right\|_2^2 \\
=&\frac{4\ell^2}{\mu^2}\delta_k \leq 4\varepsilon^2,
\end{aligned}
$$

where the first equality is based on Lemma 1, the second inequality comes from Corollary 1 and the last inequality is due to Lemma 10.

By using Jensen's inequality, we have

$$
\left(\mathbb{E}\left\|\nabla\Phi(\mathbf{x}_k) - \nabla_{\mathbf{x}} f(\mathbf{x}_k, \mathbf{y}_k)\right\|_2\right)^2 \leq \mathbb{E}\left\|\nabla\Phi(\mathbf{x}_k) - \nabla_{\mathbf{x}} f(\mathbf{x}_k, \mathbf{y}_k)\right\|_2^2 \leq 4\varepsilon^2,
$$

which means

$$
\begin{aligned}
\mathbb{E}\left\|\nabla\Phi(\mathbf{x}_k)\right\|_2 =&\mathbb{E}\left\|\nabla_{\mathbf{x}} f(\mathbf{x}_k, \mathbf{y}_k) - (\nabla_{\mathbf{x}} f(\mathbf{x}_k, \mathbf{y}_k) - \nabla\Phi(\mathbf{x}_k))\right\|_2 \\
\leq&\mathbb{E}\left\|\nabla_{\mathbf{x}} f(\mathbf{x}_k, \mathbf{y}_k)\right\|_2 + \mathbb{E}\left\|\nabla_{\mathbf{x}} f(\mathbf{x}_k, \mathbf{y}_k) - \nabla\Phi(\mathbf{x}_k)\right\|_2 \\
\leq&\mathbb{E}\left\|\nabla_{\mathbf{x}} f(\mathbf{x}_k, \mathbf{y}_k)\right\|_2 + 2\varepsilon.
\end{aligned} \tag{21}
$$

Similarly, we can use Jensen's inequality and Lemma 10 to prove

$$
\left(\mathbb{E}\left\|\mathbf{v}_k - \nabla_{\mathbf{x}} f(\mathbf{x}_k, \mathbf{y}_k)\right\|_2\right)^2 \leq \mathbb{E}\left\|\mathbf{v}_k - \nabla_{\mathbf{x}} f(\mathbf{x}_k, \mathbf{y}_k)\right\|_2^2 \leq \Delta_k \leq \frac{19}{1125}\kappa^{-2}\varepsilon^2,
$$

and

$$
\begin{aligned}
\mathbb{E}\left\|\nabla_{\mathbf{x}} f(\mathbf{x}_k, \mathbf{y}_k)\right\|_2 =&\mathbb{E}\left\|\mathbf{v}_k - (\mathbf{v}_k - \nabla_{\mathbf{x}} f(\mathbf{x}_k, \mathbf{y}_k))\right\|_2 \\
\leq&\mathbb{E}\left\|\mathbf{v}_k\right\|_2 + \mathbb{E}\left\|\mathbf{v}_k - \nabla_{\mathbf{x}} f(\mathbf{x}_k, \mathbf{y}_k)\right\|_2 \\
\leq&\mathbb{E}\left\|\mathbf{v}_k\right\|_2 + \sqrt{\frac{19}{1125}}\kappa^{-1}\varepsilon \\
\leq&\mathbb{E}\left\|\mathbf{v}_k\right\|_2 + \frac{1}{7}\varepsilon.
\end{aligned} \tag{22}
$$

By combining the inequalities (21) and (22), we obtain

$$
\begin{aligned}
\mathbb{E}\left\|\nabla\Phi(\mathbf{x}_k)\right\|_2 \leq&\mathbb{E}\left\|\nabla_{\mathbf{x}} f(\mathbf{x}_k, \mathbf{y}_k)\right\|_2 + 2\varepsilon \\
\leq&\mathbb{E}\left\|\mathbf{v}_k\right\|_2 + \frac{1}{7}\varepsilon + 2\varepsilon \\
\leq&\mathbb{E}\left\|\mathbf{v}_k\right\|_2 + \frac{15}{7}\varepsilon.
\end{aligned}
$$

$\square$

Now we can present the proof of Theorem 1.

*Proof.* Based on the update of $\mathbf{x}_k$ in Algorithm 3, we have

$$
\begin{aligned}
A_k \leq& -\eta_k\langle\nabla_{\mathbf{x}} f(\mathbf{x}_k, \mathbf{y}_k), \mathbf{v}_k\rangle + \frac{\ell\eta_k^2}{2}\left\|\mathbf{v}_k\right\|_2^2 \\
\leq&\frac{\eta_k}{2}\left\|\nabla_{\mathbf{x}} f(\mathbf{x}_k, \mathbf{y}_k) - \mathbf{v}_k\right\|_2^2 - \left(\frac{\eta_k}{2} - \frac{\ell\eta_k^2}{2}\right)\left\|\mathbf{v}_k\right\|_2^2,
\end{aligned} \tag{23}
$$

where the first inequality is due to the average smoothness of $f$, and second comes from the Cauchy-Schwartz inequality.

The choice of step size $\eta_k$ implies that

$$
\begin{aligned}
\left(\frac{\eta_k}{2} - \frac{\ell\eta_k^2}{2}\right) \|\mathbf{v}_k\|_2^2 &\geq \frac{9\varepsilon^2}{100\kappa\ell} \min\left(\frac{\|\mathbf{v}_k\|_2}{\varepsilon}, \frac{\|\mathbf{v}_k\|_2^2}{2\varepsilon^2}\right) \\
&\geq \frac{9\varepsilon^2}{100\kappa\ell}\left(\frac{\|\mathbf{v}_k\|_2}{\varepsilon} - 2\right) \\
&= \frac{9}{100\kappa\ell}\left(\varepsilon\|\mathbf{v}_k\|_2 - 2\varepsilon^2\right),
\end{aligned}
\tag{24}
$$

where the first inequality is based on $\kappa \geq 1$ and the definition of $\eta_k$; the second one uses the fact that $\min(|x|, \frac{x^2}{2}) \geq |x| - 2$ holds for all $x$.

By combining inequalities (23), (24) and taking expectation, we obtain the upper bound of $\mathbb{E}[A_k]$:

$$
\begin{aligned}
\mathbb{E}[A_k] &\leq \frac{1}{20\kappa\ell}\mathbb{E}\|\nabla_{\mathbf{x}}f(\mathbf{x}_k, \mathbf{y}_k) - \mathbf{v}_k\|_2^2 - \frac{9}{100\kappa\ell}\left(\varepsilon\mathbb{E}\|\mathbf{v}_k\|_2 - 2\varepsilon^2\right) \\
&\leq \frac{1}{20\kappa\ell}\Delta_k - \frac{9}{100\kappa\ell}\left(\varepsilon\mathbb{E}\|\mathbf{v}_k\|_2 - 2\varepsilon^2\right).
\end{aligned}
\tag{25}
$$

The definition of $\Phi^*$ and Assumption 1 implies

$$
\Phi^* - f(\mathbf{x}_K, \mathbf{y}_K) \leq f(\mathbf{x}_K, \mathbf{y}^*(\mathbf{x}_K)) - f(\mathbf{x}_K, \mathbf{y}_K) \leq \frac{134\varepsilon^2}{\kappa\ell}.
\tag{26}
$$

where the second inequality can be shown by following the proof of Lemma 14[4].

By combining inequalities (25), (26), Lemma 14 and Corollary 3; and taking the average over $k = 0, \ldots, K-1$, we obtain

$$
\frac{1}{K}\sum_{k=0}^{K-1}\mathbb{E}[f(\mathbf{x}_{k+1}, \mathbf{y}_{k+1}) - f(\mathbf{x}_k, \mathbf{y}_k)] \leq \frac{1}{K}\sum_{k=0}^{K-1}\left(\frac{1}{20\kappa\ell}\Delta_k - \frac{9}{100\kappa\ell}\left(\varepsilon\mathbb{E}\|\mathbf{v}_k\|_2 - 2\varepsilon^2\right) + \frac{134\varepsilon^2}{\kappa\ell}\right).
$$

Consequently, we have

$$
\begin{aligned}
&\frac{9\varepsilon}{100\kappa\ell}\cdot\frac{1}{K}\sum_{k=0}^{K-1}\mathbb{E}\|\mathbf{v}_k\|_2 \\
&\leq \frac{1}{K}\sum_{k=0}^{K-1}\left(\frac{1}{20\kappa\ell}\Delta_k + \frac{9\varepsilon^2}{50\kappa\ell} + \frac{134\varepsilon^2}{\kappa\ell}\right) + \frac{1}{K}\Big(f(\mathbf{x}_0, \mathbf{y}_0) - \mathbb{E}[f(\mathbf{x}_K, \mathbf{y}_K)]\Big) \\
&\leq \frac{135\varepsilon^2}{\kappa\ell} + \frac{1}{K}\Big(f(\mathbf{x}_0, \mathbf{y}_0) + \frac{134\varepsilon^2}{\kappa\ell} - \Phi^*\Big),
\end{aligned}
$$

where the second inequality uses Corollary 3 to bound $\Delta_k$.

Rearranging above result, we achieve

$$
\frac{1}{K}\sum_{k=0}^{K-1}\mathbb{E}\|\mathbf{v}_k\|_2 \leq 1500\varepsilon + \frac{100\kappa\ell}{9K\varepsilon}\Big(\mathbb{E}[f(\mathbf{x}_0, \mathbf{y}_0)] + \frac{134\varepsilon^2}{\kappa\ell} - \Phi^*\Big) = 1500\varepsilon + \frac{100\kappa\ell\Delta_f}{9K\varepsilon}.
\tag{27}
$$

According to $K = \left\lceil 100\kappa\ell\varepsilon^{-2}\Delta_f/9 \right\rceil$ and inequality (27), we have

$$
\mathbb{E}\|\nabla\Phi(\hat{\mathbf{x}})\|_2 = \frac{1}{K}\sum_{k=0}^{K-1}\mathbb{E}\|\nabla\Phi(\mathbf{x}_k)\|_2 \leq \frac{1}{K}\sum_{k=0}^{K-1}\left(\mathbb{E}\|\mathbf{v}_k\|_2 + \frac{15}{7}\varepsilon\right) = 1504\varepsilon.
$$

$\square$

## E The proof of Theorem 2

In the finite-sum case, we use the full gradient to replace the large batch sample size in stochastic case. Similar to previous section, we extend SARAH [30] to constrained case as the initialization of $\mathbf{y}_0$. We can prove Theorem 2 with minor modifications on the analysis of Theorem 1.

**Algorithm 7** PSARAH $(g(\cdot),\ K_0)$

---

1: **Input** $\mathbf{w}_0 \in \mathcal{C}$, learning rate $\gamma > 0$, inner loop size $m$
2: **for** $k = 1, \ldots, K_0$ **do**
3: $\quad \tilde{\mathbf{w}}_{k,0} = \mathbf{w}_{s-1}$
4: $\quad \tilde{\mathbf{v}}_{k,0} = \nabla g(\tilde{\mathbf{w}}_{k,0})$
5: $\quad \tilde{\mathbf{w}}_{k,1} = \Pi_{\mathcal{C}}\left(\tilde{\mathbf{w}}_{k,0} - \gamma \tilde{\mathbf{v}}_{k,0}\right)$
6: $\quad$ **for** $t = 1, \ldots, m - 1$ **do**
7: $\qquad$ draw sample $\boldsymbol{\xi}_t$
8: $\qquad \tilde{\mathbf{v}}_{k,t} = \tilde{\mathbf{v}}_{k,t-1} + \nabla G(\tilde{\mathbf{w}}_{k,t}; \boldsymbol{\xi}_t) - \nabla G(\tilde{\mathbf{w}}_{k,t-1}; \boldsymbol{\xi}_t)$
9: $\qquad \tilde{\mathbf{w}}_{k,t+1} = \Pi_{\mathcal{C}}(\tilde{\mathbf{w}}_{k,t} - \gamma \tilde{\mathbf{v}}_{k,t})$
10: $\quad$ **end for**
11: $\quad \mathbf{w}_{k+1} = \tilde{\mathbf{w}}_{k,s_k}$, where $s_k$ is uniformly sampled from $\{1, \ldots, m\}$
12: **end for**
13: **Output**: $\mathbf{w}_{K_0}$

---

### E.1 Initialization by Projected SARAH

We present the detailed procedure of projected SARAH (PSARAH) in Algorithm 7, which is used to initialize $\mathbf{y}_0$ in SREDA for problem (3) (line 2 of Algorithm 5). The algorithm considers the following convex optimization problem

$$\min_{\mathbf{w} \in \mathcal{C}} g(\mathbf{w}) \triangleq \frac{1}{n} \sum_{i=1}^{n} G_i(\mathbf{w}; \boldsymbol{\xi}_i), \tag{28}$$

where $H$ is average $\ell$-Lipschitz gradient and convex, $h$ is $\mu$-strongly convex, and $\boldsymbol{\xi}_i$ is a random vector. We have the following convergence result by using SARAH to solve problem (28).

**Theorem 4.** *For Algorithm 6 with*

$$K_0 = \left\lceil \frac{\log\left(\|\mathcal{G}_\gamma(\mathbf{w}_0)\|_2^2\right)}{\log 2} \right\rceil, m = \lceil 256\kappa \rceil \ \ and \ \ \lambda = \frac{1}{8\ell}.$$

*Proof.* By following the proof of Corollary 3 with $\tilde{\mathbf{v}}_{k,0} = \nabla g(\tilde{\mathbf{w}}_{k,0})$, we have

$$\mathbb{E}\|\mathcal{G}_\gamma(\mathbf{w}_{k+1})\|^2 \leq \frac{64\ell}{m\mu}\mathbb{E}\|\mathcal{G}_\gamma(\mathbf{w}_k)\|_2^2 + 8\ell^2 \mathbb{E}\|\tilde{\mathbf{w}}_{k,1} - \tilde{\mathbf{w}}_{k,0}\|_2^2$$

$$\leq \left(\frac{64\ell}{m\mu} + \frac{1}{4}\right)\mathbb{E}\|\mathcal{G}_\gamma(\mathbf{w}_k)\|_2^2$$

$$\leq \frac{1}{2}\mathbb{E}\|\mathcal{G}_\gamma(\mathbf{w}_k)\|_2^2.$$

Hence, we have $\mathbb{E}\|\mathcal{G}_\gamma(\mathbf{w}_{K_0})\|_2^2 \leq \zeta$. $\qquad\qquad \square$

Similar to stochastic case, we directly obtain the following result.

**Corollary 5.** *Under assumptions of Theorem 4, we can obtain $\mathbb{E}\|\mathcal{G}_\gamma(\mathbf{w}_{K_0})\|_2^2 \leq \kappa^{-2}\varepsilon^2$ with $\mathcal{O}\left((n + \kappa)\log(\kappa/\varepsilon)\right)$ stochastic gradient evaluations.*

*Proof.* Using Theorem 4 with $\zeta = \kappa^{-2}\varepsilon^2$, we have $\mathbb{E}\|\mathcal{G}_\gamma(\mathbf{w}_{K_0})\|_2^2 \leq \kappa^{-2}\varepsilon^2$. The total number of stochastic gradient evaluation is

$$K_0 \cdot (n + m)$$

$$= \left\lceil \frac{\log\left(\kappa^2\varepsilon^{-2}\|\mathcal{G}_\gamma(\mathbf{w}_0)\|_2^2\right)}{\log 2} \right\rceil \cdot (n + \lceil 256\kappa \rceil)$$

$$= \mathcal{O}\left((n + \kappa)\log(\kappa/\varepsilon)\right).$$

$\qquad\qquad \square$

## E.2   The case of $n \geq \kappa^2$

We set the parameters

$$\zeta = \kappa^{-2}\varepsilon^2, \ \eta_k = \min\left(\frac{\varepsilon}{5\kappa\ell\|\mathbf{v}_k\|_2}, \frac{1}{10\kappa\ell}\right), \ \lambda = \frac{1}{8\ell}, \ q = \lceil\kappa^{-1}n^{1/2}\rceil,$$

$$S_2 = \left\lceil\frac{3687}{76}\kappa q\right\rceil, \ K = \left\lceil\frac{100\kappa\ell\varepsilon^{-2}\Delta_f}{9}\right\rceil, \ \text{and } m = \lceil 1024\kappa\rceil.$$

Then the quantity $\Delta_{k_0}$ is zero for any $k_0$ with $\mod(k_0, q) = 0$. We can follow all analysis of Theorem 1. Note that the different values of $q$ and $\Delta_{k_0}$ do not affect the proof of Lemma 13. Therefore we still obtain $\mathbb{E}\|\nabla\Phi(\hat{\mathbf{x}})\|_2 \leq 1504\varepsilon$ by the parameters setting above. The total complexity of stochastic gradient evaluation is

$$\mathcal{O}((n+\kappa)\log(\kappa/\varepsilon)) \ + \ \mathcal{O}\left(\frac{K}{q}\cdot n\right) \ + \ \mathcal{O}(K\cdot S_2\cdot m)$$

$$=\mathcal{O}((n+\kappa)\log(\kappa/\varepsilon)) \ + \ \mathcal{O}\left(\frac{\kappa\varepsilon^{-2}}{\kappa^{-1}n^{1/2}}\cdot n\right) \ + \ \mathcal{O}\left(\kappa\varepsilon^{-2}\cdot n^{1/2}\cdot\kappa\right)$$

$$=\mathcal{O}\left(n\log(\kappa/\varepsilon) + \kappa^2 n^{1/2}\varepsilon^{-2}\right).$$

## E.3   The case of $n \leq \kappa^2$

We set the parameters

$$\zeta = \kappa^{-2}\varepsilon^2, \ \eta_k = \min\left(\frac{\varepsilon}{5\kappa\ell\|\mathbf{v}_k\|_2}, \frac{1}{10\kappa\ell}\right), \ \lambda = \frac{2}{8\ell}, \ q = 1,$$

$$S_2 = 1, \ K = \left\lceil\frac{100\kappa\ell\varepsilon^{-2}\Delta_f}{9}\right\rceil, \ \text{and } m = \lceil 1024\kappa\rceil.$$

The procedure of the algorithm means $\Delta_k = 0$ holds for all $k$ since $q = 1$. Everything is identical to Theorem 1 until Lemma 13. Then we revisit the derivation of Corollary 3. Since we have $\Delta_k = 0$, the inequalities (13) and (14) will be tighter. Hence the all original bounds still hold. The remains could still follow the proof of Theorem 1 and we finally obtain $\mathbb{E}\|\nabla\Phi(\hat{\mathbf{x}})\|_2 \leq 1504\varepsilon$.

The total complexity of stochastic gradient evaluation is

$$\mathcal{O}((n+\kappa)\log(\kappa/\varepsilon)) \ + \ \mathcal{O}\left(\frac{K}{q}\cdot n\right) \ + \ \mathcal{O}(K\cdot S_2\cdot m)$$

$$=\mathcal{O}((n+\kappa)\log(\kappa/\varepsilon)) \ + \ \mathcal{O}\left(\frac{\kappa\varepsilon^{-2}}{1}\cdot n\right) \ + \ \mathcal{O}\left(\kappa\varepsilon^{-2}\cdot 1\cdot\kappa\right)$$

$$=\mathcal{O}\left((\kappa^2 + \kappa n)\varepsilon^{-2}\right).$$