[Reviews · NeurIPS 2020]

Review 1

Summary and Contributions: This paper proposes a new algorithm called SREDA to solve a class of stochastic nonconvex-strongly-concave minimax problems. The authors propose a double loop algorithm where the inner loop solves the concave max problem with stochastic variance-reduction gradient-type method. The main component of this algorithm is the use of SARAH estimator (a biased recursive stochastic estimator) introduced in [26]. The main contribution of the paper is to show that the new algorithm can achieve O(kappa^3/epsilon^-3) oracle complexity to achieve an \epsilon-stationary point.

Strengths: -- The algorithm addresses both primal and dual variables x and y in a stochastic manner. -- The oracle complexity bound O(kappa^3/epsilon^-3) for a class of nonconvex-strongly-concave problem seems to be new for general expectation problem.

Weaknesses: Though the reviewer finds that the paper has an interesting result on the oracle complexity, it has some weaknesses as follows: -- First, the originality of the idea is not novel. Since the problem is strongly concave, the function \phi(x) defined by (2) is L-smooth, and one can apply existing techniques from stochastic nonconvex optimization to solve (2), including SARAH-type methods. So, the complexity of O(\kappa^3*epsilon^-3) is not surprised. Consequently, the novelty and theoretical contribution of this paper is incremental. -- Second, the algorithm (SREDA) in fact has three loops, where Algorithm 3 has two loops (though it is presented in one single loop), and Algorithm 4 has one loop. This structure is certainly inefficient since there are many parameters needed to be tuned. These parameters are often set based on worst-case bounds from convergence analysis or heuristically chosen. In the reviewer's opinion, SREDA is therefore not an efficient algorithm in general though it may have some advantages in particular examples. -- Third, SARAH has been extensively used in many recent papers. In the strongly convex case, it has linear rate, while in the nonconvex case, it achieves optimal complexity bounds. So, it is reasonable to achieve the result as shown in the paper. Unfortunately, the authors have not clearly state their new contribution in contrast to existing works, including compositional models as in (https://arxiv.org/abs/1908.11468). -- Fourth, the complexity results stated in Theorems 1 and 2 only hold for particular batch-sizes and parameter configuration. This is certainly very limited and challenging to set in practice.

Correctness: The reviewer is unfortunately unable to check the proof due to the lack of time, but still believes its correctness due to the reasonable complexity results shown in the paper.

Clarity: The paper is relatively well-written but it does not contain a solid experiment part.

Relation to Prior Work: The authors provide a relatively sufficient review of existing work.

Reproducibility: Yes

Additional Feedback: The reviewer would like to make the following suggestions: -- As mentioned, the algorithm has three loops, making it inefficient, It is worth to see if one can reduce to a two-loop variant by performing only one iteration of the maximization procedure. -- Since the complexity results only hold for certain setting and configuration, it is also interesting to see if the algorithm remains working for single sample or other parameter configurations to increase its flexibility. -- Relations to compositional optimization such as (https://arxiv.org/abs/1908.11468) should be considered, since they have some similarity. ####################### Updated: After reading again the paper and the response, I agree that there is some technical drawbacks when going from the compositional form as in [Zhang&Xiao] to the general minimax form considered here. Note that the compositional form can be considered as special case of the minimax form by using Fenchel conjugate. I still believe that the technical drawback can be addressed similarly using the techniques, e.g, in [24], even the author used a slightly different path. The authors should also clarify this difference and whether or not the idea in [24] can be used. I also agree with other reviewers that the novelty of this paper is limited and mostly on the technical side. However, I am happy to raise my score since it seems that this paper is one of the first works providing the best-known complexity for this problem under given assumptions.


Review 2

Summary and Contributions: The paper proposes an algorithm for solving stochastic non-convex-strongly-concave min-max optimization problems. It uses variance reduction techniques to bring down the iteration complexity of previous algorithms.

Strengths: The paper is well-written and the results are clear.

Weaknesses: The reviewer has the following concerns/questions/suggestions: - In the abstract, it is mentioned that the dependence on epsilon is optimal. Why is this true? What is the lower-bound the paper refer to? - Currently the way that the paper is written does not highlight the novelties of the work. It seems the work combines existing methods for solving min-max with exiting variance reduction modules inside. Please clarify the novelties in the analysis further (the explanation in section 5 does not highlight the difficulties and challenges). - Please explain the problem formulation in the numerical experiments section. Make the section self-contained. - As minor remarks: --- In Theorem 2, the paper can unify both cases for simplicity of presentation. For example by setting q = max {kappa^-1 n^1/2, 1}, etc. --- The following papers study the same problem instance (although non-stochastic): + Ostrovskii, D. M., Lowy, A., & Razaviyayn, M. (2020). Efficient search of first-order nash equilibria in nonconvex-concave smooth min-max problems. arXiv preprint arXiv:2002.07919. + Zhao, R. (2020). A Primal Dual Smoothing Framework for Max-Structured Nonconvex Optimization. arXiv preprint arXiv:2003.04375. + Barazandeh, B., & Razaviyayn, M. (2020, May). Solving Non-Convex Non-Differentiable Min-Max Games Using Proximal Gradient Method. In ICASSP 2020-2020 IEEE International Conference on Acoustics, Speech and Signal Processing (ICASSP) (pp. 3162-3166). IEEE. ====== I read the authors' rebuttal. I think the paper is above acceptance threshold and thus I maintain my score. However, I strongly suggest the authors to make the optimality claim rigorous. More precisely, they need to show that the hard instance in [5] lies in the optimization class they consider (i.e. Assumptions 1-5 are satisfied for the hard problem in [5]).

Correctness: The result seem to be correct.

Clarity: The paper is well-written.

Relation to Prior Work: Please see the previous comment.

Reproducibility: Yes

Additional Feedback:


Review 3

Summary and Contributions: This paper studies a nonconvex-strongly-concave minimax optimization problem and proposes a method called SREDA, which estimates the gradients more efficiently using variance reduction. The main contribution of this paper is the best known stochastic gradient complexity of O(\kappa^3\eps^{-3}), which is optimal in \eps for this setting.

Strengths: 1. It is good to have an improved bound for stochastic minimax algorithms. I believe the NeurIPS community will benefit from it. 2. I briefly check the proof and believe that most of the derivations are correct. This is a solid paper.

Weaknesses: 1. The algorithm is too complicated and I doubt if it is indeed practical when applied for solving some real application problems, e.g., adversarial learning with deep neural networks. The parameter tuning can be an issue. Current numerical results on distributionally robust logistic regression are encouraging yet not sufficiently convincing. 2. The algorithm is a straightforward combination of SARAH and SGDA with normalized gradient for x-side. The proof is seemingly largely based on the techniques used for analyzing SARAH and SPIDER.

Correctness: Yes, they are all correct.

Clarity: Yes, the paper is well written.

Relation to Prior Work: Yes, the difference is clearly discussed.

Reproducibility: Yes

Additional Feedback: This paper seems a weak accept to me. A new upper bound is welcome and has its own values. Please see the detailed comments as follows, 1. Different from the algorithms [25] which are only of theoretical interests, the variance reduction algorithms are well known for their simplicity and ease-of-the-implementation. Thus, I am wondering whether the algorithmic scheme is necessarily complicated or not. While Section 4.1 is helpful, the authors might need to provide more comments. 2. Numerical results on some deep learning application problems seem important. Indeed, as mentioned before, there are too many parameters in the algorithm and current results on LIBSVM datasets are encouraging yet not very convincing. 3. Some important references are missing, especially (Arxiv 2002.05309). This new references propose a new epoch-GDA, which improves SGDA in [24] by addressing an issue of large minibatch. 4. Please provide some remarks on the novelty of the technical contribution. To be more specific, why is nontrivial to track the gradient estimator recursively under the minimax setting? The current complicated algorithmic scheme seems to be a rather straighforward composition of standard algorithms. I am happy to raise my score if the authors address the above concerns. =========================================================== I have read the rebuttal from the authors and agree that the techniques have the novelty. However, it seems not enough to justify the effiectiveness of parameter tuning using LIBSVM datasets given the complicated scheme of the algorithm. This prevents me from raising 6 to 7. Nonetheless, I appreciate the theoretical value of the algorithm and decide to maintain my score.


Review 4

Summary and Contributions: This paper considers a nonconvex strongly concave minimax optimization problem. The authors propose variance reduction algorithms to solve this problem under the stochastic setting and the finite-sum setting. The proposed algorithms achieve better sample complexities than previous works. The authors further conduct experiments to show that their algorithms empirically have faster convergence than the existing methods.

Strengths: Novel variance reduction algorithms are proposed to solve the nonconvex strongly concave optimization under the stochastic setting and the finite-sum setting. Theoretically, the authors prove the best known sample complexity for both settings comparing to the previous works.

Weaknesses: Algorithm 4 also takes v_k, u_k, x_k, y_k and x_{k+1} as input arguments. It would be more clear if the authors add those arguments in the expression of the function ConcaveMaximizer. The major concern about this paper is the claim that O(epsilon^{-3}) is the optimal rate for the minimax problem according to [5]. Although [5] gives the lower bound of stochastic nonconvex optiminization, it is based on the stochastic first-order oracle, which is equivalent to the oracle for $\nabla_x F(x, y^*(x); \xi) $ for the minimax problem in this submission. However, the proposed algorithms are based on the oracle for $\nabla_x F(x, y; \xi)$ and $\nabla_y F(x, y; \xi)$. Is there a rigorous way to building a connection between those two oracles to show the lower bound of the nonconvex strongly concave minimax problem? ====================== Thank the authors for the response. The authors have addressed my concern about the claim of optimality. Therefore, I raise the score from 6 to 7.

Correctness: The theoretical claims look correct. The detailed proofs are not checked carefully. The empirical methodology is correct.

Clarity: This paper is well written.

Relation to Prior Work: The authors have clearly compared their results with previous works.

Reproducibility: Yes

Additional Feedback:

[Author Response · NeurIPS 2020]

**Reply to Reviewer 2, 3 and 4**

Novelty of the analysis: Besides keeping the estimator of $\nabla f(\mathbf{x}_k, \mathbf{y}_k)$ sufficiently accurate like SARAH/SPIDER for convex/nonconvex minimization, the minimax problem also requires $\mathbf{y}_k$ to be close to $\mathbf{y}^*(\mathbf{x}_k)$, which leads to a more challenging analysis. Our analysis is based on a recursive relationship between $\Delta_k = \mathbb{E}[\|\mathbf{v}_k - \nabla_{\mathbf{x}} f(\mathbf{x}_k, \mathbf{y}_k)\|_2^2 + \|\mathbf{u}_k - \nabla_{\mathbf{y}} f(\mathbf{x}_k, \mathbf{y}_k)\|_2^2]$ and $\delta_k = \mathbb{E}\|\mathcal{G}_{\lambda, \mathbf{y}}(\mathbf{x}_k, \mathbf{y}_k)\|_2^2$ (defined in line 364 of Appendix B) as in Lemma 13 (Corollary 2, line 418, Appendix B), which guarantees both $\Delta_k$ and $\delta_k$ to be smaller than $\mathcal{O}(\kappa^{-2}\varepsilon^2)$ (line 434-435, Appendix B). This is a nontrivial ingredient of the analysis. To achieve the desired convergence rate, all of the stepsizes, mini-batch sizes, number of the inner iterations, and the orders of $\Delta_k$ and $\delta_k$ must be balanced carefully. In comparison, SARAH/SPIDER for convex/nonconvex minimization only needs to consider the estimator of gradient, which does not involve the extra complexity in minimax problems.

**Reply to Reviewer 2 and 4**

Algorithm is complicated: As mentioned in the setting of experiments (line 589, Appendix F), we can select $q = m = \lceil n/S_2 \rceil$ heuristically and the empirical result show it performs well in practice. We agree that it is worth to see weather there exists a simpler variant of SREDA which also holds the theoretical guarantee.

**Reply to Reviewer 3 and 5**

Optimal dependency on $\varepsilon$: The lower bound means any stochastic first-order algorithm requires at least $\mathcal{O}(\varepsilon^{-3})$ calls of stochastic first-order oracle $(\nabla_{\mathbf{x}} F(\mathbf{x}, \mathbf{y}; \xi), \nabla_{\mathbf{y}} F(\mathbf{x}, \mathbf{y}; \xi))$ to find $\varepsilon$-stationary point of $\Phi(\mathbf{x}) = \arg\max_{\mathbf{y} \in \mathcal{Y}} f(\mathbf{x}, \mathbf{y})$ (line 93, Definition 1). As we mentioned in line 180-185, we can consider the special case of minimax problem whose objective function has the form $f(\mathbf{x}, \mathbf{y}) = g(\mathbf{x}) + h(\mathbf{y})$ where $g$ is possibly nonconvex and $h$ is strongly-concave, which leads to minimizing on $\mathbf{x}$ and maximizing on $\mathbf{y}$ are independently. Consequently, finding $\mathcal{O}(\varepsilon)$-stationary point of the corresponding $\Phi(\mathbf{x})$ can be reduced to finding $\mathcal{O}(\varepsilon)$-stationary point of nonconvex function $g(\mathbf{x})$, which is based on the stochastic first order-oracle $\nabla_{\mathbf{x}} F(\mathbf{x}, \mathbf{y}; \xi) = \nabla g(\mathbf{x}; \xi)$ (this equality holds for any $\mathbf{y}$ and we also have $\nabla_{\mathbf{x}} F(\mathbf{x}, \mathbf{y}^*(\mathbf{x}); \xi) = \nabla g(\mathbf{x}; \xi)$). Hence, the analysis of stochastic nonconvex miminization problem [5] based on $\nabla g(\mathbf{x}; \xi)$ can directly lead to the $\mathcal{O}(\varepsilon^{-3})$ lower bound for our minimax problem.

**Reply to Reviewer 2**

Relations to compositional optimization: We thank the reviewer for pointing out this valuable reference (arXiv:1908.11468). We are happy to cite this paper and compare it with SREDA. Both this work and SREDA use variance reduction to address nonconvex multi-level (two-level) optimization problem, however, their settings are quite different. We can reformulate our minimax problem (2) as compositional problem:

$$\min_{\mathbf{x}, \mathbf{y}} f_2(f_1(\mathbf{x}, \mathbf{y})) + \Psi(\mathbf{y}), \tag{a}$$

where $f_1(\mathbf{x}, \mathbf{y}) = (\mathbf{x}, \arg\max_{\mathbf{y} \in \mathcal{Y}} f(\mathbf{x}, \mathbf{y}))$, $f_2(\mathbf{x}, \mathbf{y}) = f(\mathbf{x}, \mathbf{y})$ and $\Psi(\mathbf{y})$ is the indicator function of $\mathcal{Y}$. The two-level nested-SPIDER (arXiv:1908.11468) requires to access the stochastic gradient of $f_1$ and $f_2$ to solve the compositional problem. For the two-level formulation (a) of our minimax problem, it is not natural to access the (stochastic) gradient of $f_1$ as an oracle since we can not provide the closed form of $\arg\max_{\mathbf{y} \in \mathbb{R}} f(\mathbf{x}, \mathbf{y})$ and its (stochastic) gradient in general. A more reasonable way is to solve it by accessing the (stochastic) gradient of $f$ like SREDA.

**Reply to Reviewer 3**

1. Problem formulation in experiments: Due to the space limitation, we gave the problem formulation in Appendix F. We are happy to follow the reviewer's suggestion and include it in the main text if the paper is accepted.

2. Minor remarks: Thanks for the suggestion. We will simplify the presentation of Theorem 2 and cite the papers about non-stochastic algorithms for minimax problems.

**Reply to Reviewer 4**

1 & 2. Please see "Reply to Reviewer 2, 3 and 4" and "Reply to Reviewer 2 and 4" above.

3. We thank the reviewer for pointing out this valuable reference, which is complementary to our work. We will cite it and compare to SREDA. First, the convergence result of epoch-GDA is based on the measure of nearly $\varepsilon$-stationary point because it also considers an additional constraint on $\mathbf{x}$, while SREDA is based on $\varepsilon$-stationary point. Second, the stochastic first-order oracle complexity of SREDA depends on $\mathcal{O}(\kappa^3 \varepsilon^{-3})$, while epoch-GDA depends on $\tilde{\mathcal{O}}(\kappa^2 \varepsilon^{-4})$.

**Reply to Reviewer 5**

Presentation of Algorithm 4: We thank the reviewer for this suggestion, and will add the arguments in the expression of the function ConcaveMaximizer.

[Meta-Review · NeurIPS 2020]

The reviewers all agree that the improved complexity is new for the class of stochastic nonconvex-strongly-concave minimax problems, and the claim of optimal complexity is clarified in the rebuttal. The weakness pointed out by the reviewers is the seemingly lack of novelty by combining existing techniques of SGDA and SARAH gradient estimators, but the author rebuttal is convincing that the combination in the minimax setting requires innovation. Another weakness is that the SREDA algorithm is rather complex, having multi-level loops and can be hard to tune in practice. Overall I recommend acceptance based on the value of the theoretical improvement. The authors should carefully address the remaining concerns of the reviewers in the revision, especially to clarify the claim of optimal complexity. In addition, the experiment settings should be explained more clearly in the main paper, including the problem formulation of the distributionaly robust optimization problem and hyperparameter choices.